# EXPLANATIONS OF BLACK-BOX MODELS BASED ON DIRECTIONAL FEATURE INTERACTIONS

Aria Masoomi[1]          Davin Hill[1]          Zhonghui Xu[2]          Craig P. Hersh[2]

Edwin K. Silverman[2]          Peter J. Castaldi[2]          Stratis Ioannidis[1]          Jennifer Dy[1]

## ABSTRACT

As machine learning algorithms are deployed ubiquitously to a variety of domains, it is imperative to make these often black-box models transparent. Several recent works explain black-box models by capturing the most influential features for prediction per instance; such explanation methods are univariate, as they characterize importance per feature. We extend univariate explanation to a higher-order; this enhances explainability, as bivariate methods can capture feature interactions in black-box models, represented as a directed graph. Analyzing this graph enables us to discover groups of features that are equally important (i.e., interchangeable), while the notion of directionality allows us to identify the most influential features. We apply our bivariate method on Shapley value explanations, and experimentally demonstrate the ability of directional explanations to discover feature interactions. We show the superiority of our method against state-of-the-art on CIFAR10, IMDB, Census, Divorce, Drug, and gene data.

## 1    INTRODUCTION

The ability to interpret and understand the reasoning behind black box decision-making increases user trust; it provides insights into how a model is working and, as a consequence, how a model can be improved. This has led to a large body of work on the development of explanation methods (Ribeiro et al., 2016; Chen et al., 2018; Yoon et al., 2018; Lundberg & Lee, 2017) applied to black-box models. Such methods aim to explain black-box behavior by understanding how individual features influence prediction outcomes. Recently, Covert et al. (2020a) proposed a unifying mathematical framework capturing a broad array of explainability techniques, termed *Removal-based Explanation* methods. Nevertheless, the overwhelming majority of explainability methods have a significant drawback: they only provide univariate explanations and, as a result, they do not take into account *feature interactions*. This is problematic precisely because many black box models, such as deep neural networks, perform well by creating complex structures and combining features in their latent layers. To address this, recent methods have been proposed to learn the interaction between features (Sundararajan et al., 2020a; Maas et al., 2011). Their definition of interaction assumes features affect each other symmetrically; however, in many real-world applications, feature interactions may be asymmetrical. We also observe this experimentally (see Fig. 1), and argue of the importance of developing black-box explanations that not only capture interactions, but also incorporate asymmetry. Overall, we make the following contributions:

- We propose a method to extend any given univariate removal-based explanation to a bivariate explanation model that can capture asymmetrical feature interactions, represented as a directed graph. Our method is general, and can be applied to a broad array of univariate removal-based explanations, as defined by Covert et al. (2020a).
- We show that analyzing this graph gives a semantically-rich interpretation of black boxes. In particular, beyond the ability to identify most influential features, the graph can identify *directionally redundant* features, i.e., features whose presence negates the influence of other features, as well as

---

[1]Northeastern University, Department of Electrical and Computer Engineering, Boston, MA, USA. {masoomi.a}@northeastern.edu, {dhill,ioannidis,jdy}@ece.neu.edu.

[2]Brigham and Women's Hospital, Channing Division of Network Medicine, Boston, MA, USA. {rezxu,craig.hersh,reeks,repjc}@channing.harvard.edu

*mutually redundant* features, i.e., features that are interchangeable. These two concepts cannot be captured by either univariate or symmetric bivariate explanations in existing literature.
- We systematize the analysis of the directed explanation graph, providing both formal definitions of the aforementioned notions as well as algorithms for scrutinizing and explaining black-box model behavior. We also provide theoretical justification for these definitions in the context of SHAP, the Shapley value explanation map introduced by Lundberg & Lee (2017).
- Finally, extensive experiments on MNIST, CIFAR 10, IMDB, Census, Divorce, Drug, and gene expression data show that our explanation graph outperforms prior symmetrical interaction explainers as well as univariate explainers with respect to post-hoc accuracy, AUC, and time.

## 2 RELATED WORK

Many methods have been proposed for explaining black box models (Guidotti et al., 2018). For instance, LIME (Ribeiro et al., 2016) explains the prediction of a model by learning a linear model locally around a sample, through which it quantifies feature influences on the prediction. SHAP (Lundberg & Lee, 2017) learns an influence score for each feature based on the Shapley value (Shapley, 2016). L2X (Chen et al., 2018) learns a set of most influential features per sample based on the mutual information between features and labels. Recently, Covert et al. (2020a) unified many of such explanation methods under a single framework; we present this in detail in Sec. 3.

However, all of these methods only capture the univariate influence of each feature; i.e., they do not explain feature interactions. Discovering feature interactions has drawn recent interest in machine learning (Bondell & Reich, 2008; Chormunge & Jena, 2018; Zeng & Figueiredo, 2014; Janizek et al., 2021; Zhang et al., 2021; Tsang et al., 2018; 2020). Tsang et al. (2017) proposed a framework to detect statistical interactions in a feed-forward neural network by directly interpreting the weights of the model. Cui et al. (2020) proposed a non-parametric probabilistic method to detect global interactions. However, most such methods study feature interactions globally (for all instances). In contrast, our work detects interactions per individual instance. The work more related to explainability of a black box via feature interaction is Shapley interaction. Grabisch & Roubens (1999) proposed a Shapley interaction value to explore the interaction between features rather than feature influence. Lundberg et al. (2018a) and Sundararajan et al. (2020a) applied Shapley interaction value to explain black box predictions. Instance-wise Feature Grouping (Masoomi et al., 2020) explored the effects of feature interaction by allocating features to different groups based on the similarity of their contribution to the prediction task. These methods assume a symmetrical interaction between features; in contrast, our method provides instance-wise explanations that can capture asymmetrical (directional) interactions.

Another type of explainers are Graph Neural Network (GNN) explainers (Yuan et al., 2020). These methods assume that the black-box model has a GNN architecture; i.e. the model incorporates the input graph structure in its predictions. In contrast, our method allows the black box to be any type of function (e.g., CNN, GNN, Random Forest) and does not assume access to a graph structure: we learn the feature interactions directly from the data. A small subset of GNN explainers, especially local, perturbation-based methods (such as Yuan et al. (2021); Duval & Malliaros (2021); Luo et al. (2020); Ying et al. (2019)) can be applied to black-box models. This can be done by assuming a non-informative interaction structure on the data and allowing the explainers to mask or perturb the interaction edges. However, non-GNN black box models are unable to utilize the graph structure.

Causal methods provide explanations through feature influence by utilizing knowledge about causal dependencies. Frye et al. (2020) generalized the Shapley-value framework to incorporate causality. In particular, they provided a new formulation for computing Shapley value when a partial causal understanding of data is accessible, which they called Asymmetric Shapley values. Wang et al. (2021a) extend this idea to incorporate the entire casual graph to reason about the feature influence on the output prediction. Causal methods rely on prior access to casual relationships; in contrast, our method learns the asymmetrical interaction between features rather than causal dependencies.

## 3 BACKGROUND

In general, explainability methods aim to discover the reason why a black box model makes certain predictions. In the local interpretability setting (Chen et al., 2018; Lundberg & Lee, 2017; Ribeiro et al., 2016; Sundararajan et al., 2017), which is our main focus, explanations aim to interpret predictions made by the model on an individual sample basis. Typically, this is done by attributing the

prediction to sample features that were most influential on the model's output. The latter is discovered through some form of input perturbation (Lundberg & Lee, 2017; Zeiler & Fergus, 2014). Covert et al. (2020a) proposed a framework, unifying a variety of different explainability methods. As we generalize this framework, we formally describe it below.

**Univariate Removal-Based Explanations.** The unifying framework of Covert et al. identifies three stages in a removal-based explanation method. The first, *feature-removal*, defines how the method perturbs input samples by removing features; the second, termed *model-behavior*, captures the effect that this feature removal has on the black-box model predictions; finally, the *summary* stage abstracts the impact of feature subset selection to a numerical score for each feature, capturing the overall influence of the feature in the output. Formally, a black-box model is a function $f : \mathcal{X} \to \mathcal{Y}$, mapping input features $x \in \mathcal{X} \subseteq \mathbb{R}^d$ to labels $y \in \mathcal{Y}$. Let $D \equiv \{1, \ldots, d\}$ be the feature space coordinates. Given an input $x \in \mathcal{X}$ and a subset of features $S \subseteq D$, let $x_S = [x_i]_{i \in S} \in \mathbb{R}^d$ be the projection of $x$ to the features in $S$. In the local interpretability setting, we are given a black-box model $f$, an input $x \in X$ and (in some methods) the additional ground truth label $y \in \mathcal{Y}$, and wish to interpret the output $f(x)$ produced by the model. The three stages of the removal based model are defined by a triplet of functions $(F, u, E)$, which we define below.

First, the feature removal stage is defined by a *subset function*

$$F : \mathcal{X} \times P(D) \to \mathcal{Y}, \tag{1}$$

where $P(D) = 2^D$ is the power set of $D$. Given an input $x \in \mathcal{X}$, and a set of features $S \subseteq D$, the map $F(x, S)$ indicates the label generated by the model when feature subset $S$ is given. For example, several interpretability methods (Yoon et al., 2018; Chen et al., 2018) set $F(x, S) = f([\mathbf{0}; x_S])$, i.e., replace the "removed" coordinates with zero. Other methods (Lundberg et al., 2020; Covert et al., 2020b) remove features by marginalizing them out using their conditional distribution $p(X_{\bar{S}}|X_S = x_S)$, where $\bar{S} = D \setminus S$, i.e., $F(x, S) = \mathbb{E}[f(X)|X_S = x_s]$.

Having access to the subset function $F$, the model behavior stage defines a *utility* function

$$u : P(D) \to \mathbb{R} \tag{2}$$

quantifying the utility of a subset of features $S \subseteq D$. For instance, some methods (Covert et al., 2020b; Schwab & Karlen, 2019) calculate the prediction loss between the true label $y$ for an input $x$, using a loss function $\ell$, i.e., $u(S) = -\ell(F(x, S), y)$. Other methods (Chen et al., 2018; Yoon et al., 2018) compute the expected loss for a given input $x$ using the label's conditional distribution, i.e., $u(S) = -\mathbb{E}_{p(Y|X=x)}[\ell(F(x, S), Y)]$.

The utility function can be difficult to interpret due to the exponential number of possible feature subsets. This is addressed by the summary stage as follows. Let $\mathcal{U} = \{u : P(D) \to \mathbb{R}\}$ be the set of all possible utility functions. An *explanation map* is a function

$$E : \mathcal{U} \to \mathbb{R}^d \tag{3}$$

mapping a utility function to a vector of scores, one per feature in $D$. These scores summarize each feature's value and are the final explanations produced by the removal-based explainability algorithm $(F, u, E)$. For instance, some methods (Zeiler & Fergus, 2014; Petsiuk et al., 2018; Schwab & Karlen, 2019) define $E(u)_i = u(D) - u(D \setminus \{i\})$, or $E(u)_i = u(\{i\}) - u(\emptyset)$ (Guyon & Elisseeff, 2003). Some methods learn $E(u)$ by solving an optimization problem (Chen et al., 2018; Ribeiro et al., 2016; Yoon et al., 2018). For example, L2X defines $E(u) = \underset{S:|S|=k}{\arg\max}\, u(S)$ for a given $k$.

**The Shapley Value Explanation Map.** In our experiments, we focus on Shapley value explanation maps. Shapley (Shapley, 2016) introduced the Shapley value in coalition/cooperative games as a means to compute "payouts", i.e., ways to distribute the value of a coalition to its constituent members. Treating features as players in such a coalition, the Shapley value has been used by many research works on explainability to compute feature influence (Datta et al., 2016; Lundberg et al., 2020; Covert et al., 2020b). For instance, Lundberg & Lee (2017) proposed SHAP as a unified measure of feature influence which uses Shapley value for summarization. They also showed that explainers such as LIME , DeepLIFT (Shrikumar et al., 2017), LRP (Bach et al., 2015), QII (Datta et al., 2016) can all be described by SHAP under different utility functions. Formally, given a utility function $u$, the SHAP explanation map $E(u)$ has coordinates:

$$E(u)_i = \sum_{S \subseteq D \setminus \{i\}} \frac{|S|!\,(d-|S|-1)!}{d!}(u(S \cup \{i\}) - u(S)), \tag{4}$$

where $S \in P(D)$. Direct computation of Eq. (4) is challenging, as the summands grow exponentially as the number of features increases. Štrumbelj & Kononenko (2014) proposed an approximation with Monte Carlo sampling, known as Shapley sampling values. Lundberg & Lee (2017) introduced KernelSHAP and DeepSHAP to compute the shapley values using kernel and neural network approaches respectively, and showed that such methods require fewer evaluations of the original model to obtain similar approximation accuracy as prior methods.

## 4 FROM UNIVARIATE TO MULTI-VARIATE EXPLANATIONS

Univariate removal-based explanation methods presented in the previous section share a similar limitation: they do not explain feature interactions in the black box model. Given a *univariate* explanation map $E : \mathcal{U} \to \mathbb{R}^d$, we propose a method to extend $E$ to its *Bivariate* explanation map which discovers feature interactions. Let $u \in \mathcal{U}$ be the utility function $u : P(D) \to \mathbb{R}$. Given this $u \in \mathcal{U}$, we define the bivariate explanation $E^2 : \mathcal{U} \to \mathbb{R}^{d \times d}$ as a $d \times d$ matrix: the $i^{\text{th}}$ column of this matrix is $E(u_i)$, i.e., the univariate explanation applied to utility function $u_i$, where $u_i$ is defined as:

$$u_i : P(D) \to \mathbb{R} \quad \text{s.t.} \quad \forall S \in P(D),\, u_i(S) = \begin{cases} u(S), & \text{if } i \in S, \\ 0, & \text{if } i \notin S. \end{cases} \tag{5}$$

Intuitively, the utility function $u_i$ is a restriction of $u$ to sets in which when $i$ is an included feature. As a result, $E(u_i) \in \mathbb{R}^d$ determines a feature's influence *conditioned on the presence of feature $i$.* [1] We denote the $j^{\text{th}}$ element by $E(u_i)_j = E^2(u)_{ji}$ which represents the importance of feature $j$ conditioned on feature $i$ being present.

**Bivariate Shapley Explanation Map.** As a motivating example, let explanation map $E : \mathcal{U} \to \mathbb{R}^d$ be the Shapley map , defined in Eq. (4). Applying the *Bivariate explanation* extension (5) to the Shapley value we obtain:

$$E^2(u)_{ij} = \sum_{j \in S \subseteq D \setminus \{i\}} \frac{|S|!\,(d-|S|-1)!}{d!}(u(S \cup \{i\}) - u(S)). \tag{6}$$

We provide the derivation of this formula in App. E in the supplement. An important feature of the above bivariate explanation is that it is *not symmetric*: in general, $E^2(u)_{ij} \neq E^2(u)_{ji}$. In other words, feature $i$ may influence $j$ differently than how feature $j$ influences $i$. This is in sharp contrast with other bivariate explanation methods, such as, e.g., interaction Shapley (Grabisch & Roubens, 1999; Sundararajan et al., 2020a), that are symmetric. Hence, $E^2(u)$ is an asymmetric matrix, that can be represented by a weighted directed graph denoted by $\mathcal{G} = (V_\mathcal{G}, E_\mathcal{G}, W_\mathcal{G})$, where weights $W_\mathcal{G}(i, j) = E^2(u)_{ji}$ (see App. B for a brief review of graph terminology). We call $\mathcal{G}$ the *directed explanation graph*. The directionality of $G$/asymmetry of $E^2$ has important implications for explainability, which we illustrate next with an example.

**Illustrative Example (Univariate Shapley vs. Bivariate Shapley).** In order to motivate the bivariate explanation map, highlight its difference from the univariate Shapley explanation, and illustrate the importance of directionality, we study the directed explanation graph of one sample of "The Large Movie Review Dataset (IMDB) (Maas et al., 2011)". This is a dataset of movie reviews with labels indicating positive or negative sentiment. We used a Recurrent Neural Network (RNN) as the black box and SHAP as the univariate explainer. Specifically, given a black box model $f$ and a point $x$, SHAP chooses $F(x, S) = \mathbb{E}[f(X)|X_S = x_s]$, $u = F(x, S)$, and $E$ to be the explanation map using Shapley value. We compute both the univariate Shapley explanation $E$, as well as the directed explanation graph $G$/bivariate explanation $E^2$ for the sentence, "The movie is the worst; surprisingly awful", which is predicted to have negative sentiment. Both explanations are shown in Fig. 1. We observe the following differences:

**The influence of word '*surprisingly*'**: In Fig. 1(a), we observe that SHAP explanation $E$ identifies 'awful', 'worst', 'surprisingly' as the most influential features. The negative Shapley value for 'Surprisingly' indicates that this feature affects prediction strongly in favor of a positive label (opposite of the black box outcome). However, looking at $E^2$, we realize this explanation is in fact misleading. The absence of an edge from 'worst' → 'surprisingly' suggests that *in the presence of 'worst' the word 'surprisingly' has no influence.* Interestingly, the reverse edge does exist; hence, presence of 'surprisingly' does not remove the influence of 'worst', which still influences the classification

---

[1]This approach can also be directly generalized beyond bivariate to *multivariate* explanations (see App. F).

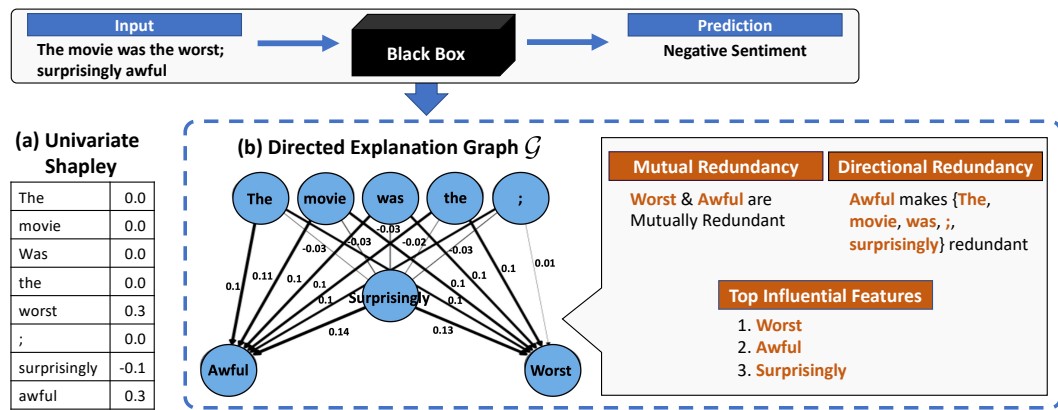

Figure 1: **a)** Univariate Shapley value. The values suggest that 'surprisingly', 'worst', 'awful' are the most influential features, however it does not explain feature interactions **b)** Using Shapley explanation map $E$, we plot the Directed Explanation graph $\mathcal{G}$ using our method. An edge $i \rightarrow j$ represents the conditional influence of word $j$ when word $i$ is present. We can then use the properties of Graph $\mathcal{G}$ to derive notions of Mutual and Directional Redundancy, as well as Influential Features.

outcome. This lack of symmetry is informative, and would not be detected from either a univariate or a bivariate but symmetric explanation.

**'awful' vs 'worst'**: The univariate explanation $E$ suggests that 'awful' and 'worst' are both influential. However, from the Shapley graph, we observe the absence of an edge from 'awful' $\rightarrow$ 'worst' and vice versa. This indicates that *the presence of either word negates the influence of the other on the classification outcome, making it redundant*. Another way to interpret this is that 'awful' and 'worst' are interchangeable. This is aligned with our understanding that awful and worst have a similar meaning, and is an observation that is not evident from the univariate SHAP $E$ alone.

**Least Influential Features**: Words like 'The', 'movie' have Shapley value zero. Such words are *sources* in the directed explanation graph $G$, i.e., have only outgoing edges towards 'awful', 'worst', and 'surprisingly' but no incoming edges at all. This suggests that they are, overall, not influential; more generally, there is a consistency between features that $E$ identifies as non-influential and *sources* in $G$.

**Most Influential Features**: The support of $E$ (i.e., words 'awful', 'worst', 'surprisingly') represents the words that have the greatest influence. In graph $G$, we observe that 'awful' and 'worst' are the *sinks* of the graph (have only incoming edges, and no outgoing edges); even though 'surprisingly' is not a sink, it still has many edges pointing into it. This is intuitive: sinks, that have no outgoing edges, indicate that all other words lose their influence when these words are present. 'Surprisingly' is important, but there are still other words that negate its influence (namely, 'awful' and 'worst').

In summary, the above example illustrates how $E^2$ and $G$ reveal more information than was present in $E$ alone. Although in a more nuanced way, the most and least influential features again stand out; however, in addition to this, features deemed as influential by SHAP can be discovered to be less influential when observed in conjunction with other features; similarly, groups of influential but mutually redundant features may also arise. Both enhance our understanding of how the black box makes predictions, and neither are observable from univariate explanations.

## 4.1 Analyzing the Directional Explanation Graph

Motivated by the above observations, we turn our attention to means of systematizing the analysis and interpretation of the Directional Explanation graph. In particular, we identify ways to discover the following notions from the graph $\mathcal{G}$: (a) the most influential features for the black box model, taking their interaction into account, and the extent to which they act as *sinks* of $\mathcal{G}$; (b) inferring redundancies between the features, by formally defining and detecting *Directional Redundancy* (e.g. 'surprisingly' and 'awful') and *Mutual Redundancy* (e.g. 'awful' and 'worst'); and (c) the transitivity properties of mutually redundant features: if mutual redundancy is transitive, it implies the existence of entire equivalence classes of features that are all interchangeable. Our goal in this section is to

both state such concepts formally, but also describe algorithms through which these concepts can be used to scrutinize and interpret $E^2$ and $\mathcal{G}$.

**Most Influential Features in Graph $\mathcal{G}$.** Most influential features in $E^2$ can be identified as sinks in $G$. These are easy to identify in polynomial time (as nodes with no outgoing edges). Nevertheless, a more graduated approach seems warranted, to identify nodes that are 'almost' sinks (like 'surprisingly' in Fig. 3). This can be accomplished through a random-walk inspired harmonic function on $\mathcal{G}$, like the classic PageRank (Page et al., 1999) or HITS (Kleinberg, 1999).[2] These correspond to the steady state distribution over a random walk on the graph with random restarts; as such it indeed offers a more graduated version of the notion of "sinkness", as desired. Variants, such as personalized PageRank (Page et al., 1999), can also be used, whereby the random restart is to a node sampled from a predetermined distribution over the vertices of $\mathcal{G}$. Setting this to be proportional to the magnitude of the univariate Shapley value $E$ interpolates between the univariate map (that captures univariate influence but not directionality) and $E^2$ (that captures directionality and "sinkness").

**Directional Redundancy and Mutual Redundancy.** In the example stated above, using a bivariate explanation map enabled us to discover which features are redundant with respect to other features. One of these examples was symmetric (e.g., 'awful' and 'worst') and one was one-sided (e.g., 'awful' makes 'surprisingly' redundant, but not vice-versa). Motivated by this, we define:

**Definition 4.1.** Given $i, j \in D$, $i$ is *directionally redundant* with respect to feature $j$ if $E^2(u)_{ij} = 0$.

Directionality arises in Def. 4.1 because $E^2(u)$ in general is not symmetric, i.e., $E^2(u)_{ij} \neq E^2(u)_{ji}$. Nevertheless, we can have features $i, j$ that have the same influence on the model (e.g. 'awful' and 'worst' in the example). We formalize this idea to the $E^2$ explanation through the following definition:

**Definition 4.2.** Given $i, j \in D$, features $i, j$ are *mutually redundant* if $E^2(u)_{ij} = E^2(u)_{ji} = 0$.

**Transitivity of Mutually Redundant Features.** Given that mutual redundancy is symmetric, it is natural to ask if it is also transitive: if 'bad' and 'awful' are mutually redundant, and so are 'awful' and 'terrible', would that imply that 'bad' and 'terrible' are also mutually redundant? This behavior is natural, and suggests that *groups of features* may act interchangeably (here, corresponding to variants of 'bad'). Identifying such groups is important for interpreting the model, as it exhibits an invariant behavior under the exchange of such features. We thus turn our attention to studying the transitivity properties of mutual redundancy. To do this, we define unweighted directed graph $\mathcal{H} = (V_{\mathcal{H}}, E_{\mathcal{H}})$, where $V_{\mathcal{H}} = V_{\mathcal{G}}$ and $E_H = \{(i, j) \in E_{\mathcal{G}} | W_{\mathcal{G}}(i, j) = 0\}$. Graph $\mathcal{H}$ captures the redundancies between any two features: an edge from $i$ to $j$ (i.e., $i \rightarrow j$) indicates that feature $j$ is directionally redundant with respect to feature $i$. We call $\mathcal{H}$ the *Redundancy Graph*. In practice, we may use the relaxed version of redundancy. Given a redundancy threshold $\gamma$, we define $\mathcal{H}_\gamma = (V_{\mathcal{H}}, E_{\mathcal{H}}^\gamma)$ to be a graph where $V_{\mathcal{H}} = V_{\mathcal{G}}$ and $E_{\mathcal{H}}^\gamma = \{(i, j) \in E_{\mathcal{G}} : |W_{\mathcal{G}}(i, j)| \leq \gamma\}$. Intuitively, if the presence of a feature makes the influence of the other less than the threshold $\gamma$, we still declare the latter to be redundant. If mutual redundancy is transitive, Graph $\mathcal{H}$ is also transitive; formally:

**Definition 4.3.** An unweighted directed graph $\mathcal{H}$ is *transitive* if $(i, j), (j, k) \in E_{\mathcal{H}}$, then $(i, k) \in E_{\mathcal{H}}$.

In other words, a transitive graph comprises a collection of cliques/fully connected graphs, captured by mutual redundancy, along with possible 'appendages' (pointing out) that are due to the non-symmetry of directed redundancy. Not every explanation graph $\mathcal{G}$ leads to a transitive $\mathcal{H}$. In the following theorem however, we prove that the Shapley explanation map $E$ indeed leads to the transitivity of mutual redundancy and, thereby, the graph $\mathcal{H}$:

**Theorem 1.** *For* $i, j, k \in D$, *assume that* $\max_{j \in S \subseteq N} |u(S \cup \{i\}) - u(S)| \leq \varepsilon_j$ *and* $\max_{i \in S \subseteq N} |u(S \cup \{k\}) - u(S)| \leq \varepsilon_i$. *Then, the following inequalities hold:* $|E^2(u)_{ij}| \leq \frac{d!}{2}\varepsilon_j$, $|E^2(u)_{ki}| \leq \frac{!}{2}\varepsilon_i$, *and* $|E^2(u)_{kj}| \leq \frac{d!}{2}(2\varepsilon_j + \varepsilon_i)$.

In short, Theorem 1 states that for a given path $i \rightarrow j \rightarrow k$ the summation of upper bounds for edges $i \rightarrow j$ and $j \rightarrow k$ can be used to upper bound the weight of the edge $i \rightarrow k$. An immediate implication of this "triangle-inequality"-like theorem is that if all $\epsilon = 0$, as is the case in directed redundancy, the weight of edge $i \rightarrow k$ must also be zero. In other words:

**Corollary 1.1.** *Graph $\mathcal{H}$ for Shapley explanation map with a monotone utility function is transitive.*

---

[2]In the context of HITS, "almost" sources and sinks correspond to "hubs" and "authorities", respectively.

Most importantly, Theorem 1 proves something stronger than that. In fact, it allows for 'almost transitivity' of graph $\mathcal{H}^\gamma$ in the case of approximate Shapley value computation: even if we set the threshold $\gamma$ to a non-zero value, short paths (of length $\gamma/\epsilon$) will also be guaranteed to be transitive. The proofs for Thm. 1 and Corollary 1.1 are provided in the supplement.

**Sources and Sinks in the Redundancy Graph.** Beyond identifying classes of mutually redundant features, the redundancy graph $\mathcal{H}$ can also be used to identify (classes of) features under which all remaining features become redundant. This can be accomplished in the following fashion. First, the strongly connected components/classes of mutually redundant features need to be discovered. As a consequence of Thm. 1, such strongly connected components will be cliques if the exact Shapley explanation map is used, or almost cliques if the threshold $\gamma$ is non-zero. Collapsing these connected components we obtain a DAG, also known as the *quotient graph* (see App B). The sources of this quotient graph (which may correspond to entire classes) correspond to feature classes that make all other features redundant (possibly through transitivity). Note the distinction between sinks in $\mathcal{G}$ and sources in $\mathcal{H}$, that may be a different set of nodes, and capture a different kind of importance. Again, rather than determining importance simply from the fact that a node in $\mathcal{H}$ is a source, a more graduated approach, using a harmonic function of nodes like Pagerank (over $\mathcal{H}$) could be used.

## 5 EXPERIMENTS

In this section, we investigate the ability of Bivariate Shapley for discovering (a) mutual redundancy, (b) directional redundancy, and (c) influential features, from black-box models. Fully evaluating Shapley values is computationally expensive; we implement Bivariate Shapley using two different Shapley approximation methods for comparison, Shapley Sampling (BivShap-S) and KernelSHAP (BivShap-K). The algorithms are outlined in App. G.1.1. The KernelSHAP approximation significantly reduces computational time (Tbl. 1) at the cost of slightly reduced Post-hoc accuracy results (Tbl. 3). In our method comparisons, we take 500 test samples from each dataset (less if the test set is smaller than 500) and generate their respective $\mathcal{G}$ and $\mathcal{H}$ graphs. We select $\gamma = 10^{-5}$ for the threshold in converting $\mathcal{G}$ to $\mathcal{H}$, which generally corresponds to 50% average graph density across the datasets (see App. G.2.4 for details). All experiments are performed on an internal cluster with Intel Xeon Gold 6132 CPUs and Nvidia Tesla V100 GPUs. All source code is publicly available.[3]

**Data.** We evaluate our methods on COPDGene (Regan et al., 2010), CIFAR10 (Krizhevsky, 2009) and MNIST (LeCun & Cortes, 2010) image data, IMDB text data, and on three tabular UCI datasets (Drug, Divorce, and Census) (Dua & Graff, 2017). We train separate black-box models for each dataset. MNIST, CIFAR10, COPDGene, and Divorce use neural network architectures; Census and Drug use tree-based models (XGBoost (Chen & Guestrin, 2016) and Random Forest). Full dataset and model details can be found in App. G.1.3.

**Competing Methods.** We compare our method against both univariate and bivariate, instance-wise black-box explanation methods. Univariate methods Shapley sampling values (Sh-Sam), KernelSHAP (kSHAP), and L2X are used to identify the top important features, either through feature ranking or by choosing a subset of features. Second-order methods Shapley Interaction Index (Sh-Int) (Owen, 1972), Shapley-Taylor Index (Sh-Tay) (Sundararajan et al., 2020b), and Shapley Excess (Sh-Exc) (Shoham & Leyton-Brown, 2008) capture symmetric interactions, on which we apply the same PageRank algorithm as Bivariate Shapley to derive a feature ranking. We also compare to a GNN explanation method GNNExplainer (GNNExp). Further details are provided in App. G.1.

### (a) Mutual Redundancy Evaluation.

We evaluate the validity of mutually redundant features through the change in model accuracy after masking redundant features, with post-hoc accuracy results shown in Fig. 2. We identify such features as groups of strongly connected nodes in graph $\mathcal{H}$, which we find using a depth-first search on $\mathcal{H}$ using Tarjan's algorithm (Tarjan, 1972). After finding the mutually redundant groups, we test their exchangeability by randomly selecting subsets of features within the group to mask. We evaluate post-hoc accuracy at different levels of masking. Masking all but one feature results in the quotient graph $\mathcal{S}$, which is represented by each dataset's marker in Fig. 2. Note that these groups can be discovered by other second-order methods by similarly interpreting the resulting feature interactions as an adjacency matrix; we include the results for applying the same algorithm to Sh-Int, Sh-Tay and

---

[3]https://github.com/davinhill/BivariateShapley

Figure 2: Post-hoc accuracy evaluated on Mutual Redundancy masking derived from graph $\mathcal{H}$. Strongly connected nodes in $\mathcal{H}$ are randomly masked with increasing mask sizes until a single node remains, represented by the final marker for each dataset. Note that we cannot run Sh-Tay and Sh-Exc on COPD due to their computational issues with large numbers of features.

| Time Complexity: Seconds per Sample | | | | | | |
|---|---|---|---|---|---|---|
| Dataset | #Features | BivShap-S | BivShap-K | Sh-Int | Sh-Tay | Sh-Exc |
| COPD | 1077 | 5942 | 36 | 2877 | 112900 | 838200 |
| CIFAR10 | 255 | 218 | 2.5 | 101 | 2819 | 6267 |
| MNIST | 196 | 116 | 1.5 | 48 | 1194 | 2350 |
| IMDB | ≤400 | 207 | 1.9 | 160 | 1279 | 1796 |
| Census | 12 | 2.7 | 0.20 | 2.6 | 11.6 | 5.3 |
| Divorce | 54 | 18.2 | 0.34 | 6.5 | 63.2 | 93.3 |
| Drug | 6 | 2.3 | 0.07 | 1.21 | 10.1 | 0.96 |

| | Post-hoc Accy | | % Feat Masked | |
|---|---|---|---|---|
| Dataset | $\mathcal{H}$-Sink Masked | $\mathcal{H}$-Source Masked | $\mathcal{H}$-Sink Masked | $\mathcal{H}$-Source Masked |
| COPD | 99.5 | 62.7 | 1.5 | 98.5 |
| CIFAR10 | 94.6 | 15.0 | 6.2 | 93.8 |
| MNIST | 100.0 | 13.4 | 77.7 | 22.3 |
| IMDB | 100.0 | 54.0 | 3.5 | 96.5 |
| Census | 100.0 | 82.0 | 23.8 | 76.2 |
| Divorce | 100.0 | 51.5 | 22.2 | 77.8 |
| Drug | 100.0 | 48.5 | 43.5 | 56.5 |

Table 1: Execution time comparison. Results are calculated on the time to produce the interaction matrix (including all features) for a single sample, as measured by seconds per sample.

Table 2: Post-hoc accuracy of BivShap-S after masking $\mathcal{H}$-source nodes, representing features with minimal redundancies, and $\mathcal{H}$-sink nodes, representing directionally redundant features.

Sh-Exc. We observe that masking an arbitrary number of mutually redundant features has minimal impact to accuracy under the BivShap method. In contrast, the groups identified by the other methods do not share the same level of feature interchangeability. In terms of finding mutual redundancy, this suggests that incorporating directionality is critical in identifying mutually redundant features; undirected methods do not capture the full context of feature interactions. We also note that no mutually redundant features were found by GNNExplainer, which indicates that the edges of its explanation graph are unable to capture feature interchangeability.

**(b) Directional Redundancy Evaluation.** We validate directional redundancy using post-hoc accuracy as shown in Tbl. 2. Additional results for BivShap-K are shown in App. G.2.3. We identify directionally redundant features as $\mathcal{H}$-sink nodes, which we identify using PageRank. We collapse the strongly connected components in graph $\mathcal{H}$ to its quotient graph $\mathcal{S}$, then apply PageRank to each connected graph in $\mathcal{S}$. The maximum and minimum ranked nodes in each connected graph correspond to the sinks and sources in $\mathcal{S}$, respectively. The condensed sinks and sources are expanded to form $\mathcal{H}$-sinks and $\mathcal{H}$-sources. We then evaluate the relevance of the identified nodes using post-hoc accuracy of masked samples. To validate our claims of directionality, we compare the results of masking $\mathcal{H}$-source nodes and $\mathcal{H}$-sink nodes, where $\mathcal{H}$-sink nodes represent the directionally redundant features. We observe in Tbl. 2 that masking sinks has little effect on accuracy, suggesting that these features contain redundant information given the unmasked source nodes. In contrast, masking $\mathcal{H}$-source nodes and keeping sinks results in large decreases in accuracy, indicating that prediction-relevant information is lost during masking.

**(c) Influential Feature Evaluation.** While mutual and directional redundancy can be investigated individually using graph $\mathcal{H}$, we can combine both concepts in a continuous ranking of features based on the steady-state distribution of a random walk on graph $\mathcal{G}$. We add a small $\epsilon \approx 0$ value to the adjacency matrix of $\mathcal{G}$ to ensure that all nodes are connected, then directly apply PageRank. The resulting feature scores are ranked, with high-scoring features representing high importance. We observe the experimental results in Fig. 3. BivShap-S consistently performs the best across data sets, including against BivShap-K, which suggests some small accuracy tradeoff for the faster approximation. Note that Bivariate Shapley does not explicitly utilize feature importance, but rather each feature's steady-state distribution. However, we can also incorporate feature importance

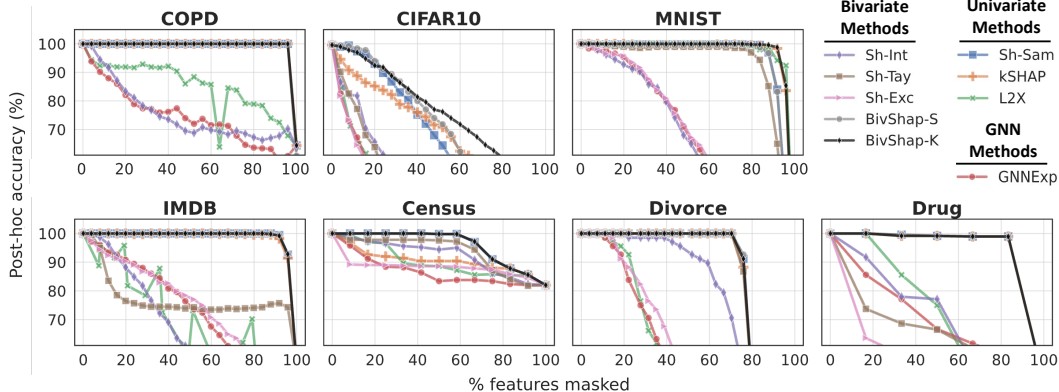

Figure 3: Comparison of explanation methods on a feature removal task. Methods are evaluated on their ability to maintain post-hoc accy while removing the least influential features. We apply PageRank to graph $\mathcal{G}$ to derive a univariate ranking based on feature redundancy. We compare to other explanation methods by iteratively masking the lowest ranked features. Note that we cannot run Sh-Tay and Sh-Exc on COPD due to their computational issues with large numbers of features.

information, defined by univariate Shapley, through Personalized PageRank. Personalization increases the steady state probabilities for important features.

**Additional Illustrative Examples from MNIST and CIFAR10 Datasets**

In Fig. 4, we investigate examples from MNIST and CIFAR10 for illustrative purposes. We see that images with homogenous background pixels show larger amounts of redundancy as identified using graph $\mathcal{H}$ in the middle row. This redundancy is also evidenced in the directional redundancy ranking in the bottom row, where the mutually redundant pixels are shown to have similar PageRank scores. We observe that pixels with higher scores make sense and correspond to pixels where the foreground object lies in the image and at key areas of the object.

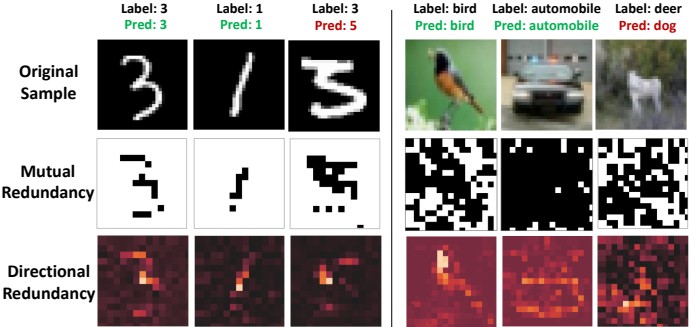

Figure 4: We explore samples from MNIST and CIFAR10. Middle row: we identify Mutually Redundant features from graph $\mathcal{H}$, indicated by the white pixels in each image. Bottom row: We apply redundancy ranking on graph $\mathcal{G}$ and show a heatmap of PageRank scores; important nodes have higher PageRank scores.

**Time Comparisons and Additional Results.** Tbl. 1 reports the time in seconds per sample to calculate feature interactions. Note that BivShap-K performs the best among Bivariate methods. Full details of timing setup is listed in App. G.2.5. In the appendix we also provide additional evaluation results based on AUC (App. G.2.1) as well as a Gene Ontology enrichment analysis of the feature rankings identified in the COPD dataset (App. G.2.6).

## 6 CONCLUSION

We extend the removal-based explanation to bivariate explanation that captures asymmetrical feature interactions per sample, which we represent as a directed graph. Using our formulation, we define two concepts of redundancy between features, denoted as mutual redundancy and directional redundancy. Our theoretical contribution leads to a systematic approach for detecting such redundancies through finding the cliques and sources in graph $\mathcal{H}$. Experiments show the benefit of capturing directional redundancies and the superiority of our method against competing methods. We discuss societal impacts in App. A.

## ACKNOWLEDGEMENTS

The work described was supported in part by Award Numbers U01 HL089897, U01 HL089856, R01 HL124233, R01 HL147326 and 2T32HL007427-41 from the National Heart, Lung, and Blood Institute, the FDA Center for Tobacco Products (CTP).

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

## APPENDIX A  SOCIETAL IMPACTS

As machine learning algorithms are widely deployed to a variety of domains such as health care, criminal justice system, and financial markets (LeCun et al., 2015; Doshi-Velez & Kim, 2017; Faghihpirayesh et al., 2021; Wang et al., 2021b; Lipton, 2018; Caruana et al., 2015; Kim et al., 2015; Wu et al., 2018), it is important to make these often black-box models transparent. Understanding the reasoning behind the decisions in black-box models may enable users' trust in the model. This paper proposes a novel method to explain black-box models. In particular, we extend univariate feature removal-based explanations to higher-order bivariate explanations that allows discovery of directional feature interactions. Explainability opens up future research directions that can help data scientists check for bias, fairness, and vulnerabilities of the models they use  (Arrieta et al., 2020; Guidotti et al., 2018).

This paper provides a general machine learning approach for explaining black-box models that can be applied to any data. We care about possible societal impact of applying machine learning to advance our understanding of disease. In this paper, our explanation method allows us to find the most influential genes for differentiating smokers versus non-smokers, potentially leading to a better understanding of the relationship between smoking and lung disease by identifying how genes interact with each other for a smoker versus a non-smoker. We plan to pursue further analysis of our results with careful guidance and insights from our medical expert collaborators. Machine learning (ML) can be applied to a variety of applications (to do harm or good), we encourage our colleagues to apply ML to beneficial applications such as health. Nevertheless, to be done correctly, one needs to work closely with domain experts to make proper judgments and lead to accurate conclusions. To increase the impact of our method, we make our source code publicly available.[4]

## APPENDIX B  GRAPH PRELIMINARY

**Directed Graph (Digraph)**:

A Directed Graph $\mathcal{G}$ is defined by a pair $(V, E)$, where $V$ is a non-empty finite set of elements called **vertices** and $E$ is a finite set of ordered pairs of distinct vertices called **arcs** or edges

**Subdigraph**:

A digraph $\mathcal{H} = (V_{\mathcal{H}}, E_{\mathcal{H}})$ is subdigraph of a digraph $\mathcal{G} = (V_{\mathcal{G}}, E_{\mathcal{G}})$ if $V_{\mathcal{H}} \subseteq V_{\mathcal{G}}$ and $E_{\mathcal{H}} \subseteq E_{\mathcal{G}}$ and every edge in $E_{\mathcal{H}}$ has both end-vertices in $V_{\mathcal{H}}$. One says $\mathcal{H}$ is **induced**  by $V_{\mathcal{H}}$ and call $\mathcal{H}$ an induced subdigraph of $\mathcal{G}$ (Bang-Jensen & Gutin, 2008).

**Degree of a Directed Graph**

Given $v \in V$ the **indegree** of $v$ is denoted as $d^-(v)$ which is the number of edges that points to $v$ and the outdegree is denoted $d^+(v)$ which is the number of edges that points out from $v$ to some other vertices. A node $v \in V$ is a source if $d^-(v) = 0$ and it is a **sink** if $d^+(v) = 0$ (Bang-Jensen & Gutin, 2008).

**Weighted Directed Graph**: It is a Directed Graph $\mathcal{G} = (V, E)$ with a mapping $W : E \to \mathbb{R}$ which assigns values to each edge. Hence, $\mathcal{G}$ can be shown as a triplet $(V, E, W)$ (Bang-Jensen & Gutin, 2008).

**Walk**:

A walk in directed graph $\mathcal{G} = (V, E)$ is an alternating sequence $W = x_1 a_1 x_2 a_2 x_3 \ldots x_{k-1} a_{k-1} x_k$ where $x_i \in V$, $1 \leq \forall i \leq k$ and $a_i \in E$ such that $a_i = (x_i, x_{i+1})$ (Bang-Jensen & Gutin, 2008).

**Strongly connected components (SCC)**

In a directed graph $\mathcal{G}$ vertex $y$ is **reachable** from vertex $x$ if there is walk from $x$ to $y$. A directed graph $\mathcal{G}$ is strongly connected if for every pair of $x, y \in V$, $x$ is reachable from $y$ and vice versa.

A strongly connected component of an directed graph $\mathcal{G}$ is a maximal induced subgraph that is strongly connected.

---

[4]https://github.com/davinhill/BivariateShapley

**Complete Graph.** A directed graph $\mathcal{G} = (V, E)$ is complete, if for every pair $x, y \in V$, we have $(x, y), (y, x) \in E$ (Bang-Jensen & Gutin, 2008).

**Cliques**: A clique is complete subdigraph of a given graph (Meeusen & Cuyvers, 1975).

**Quotient Graph $\mathcal{S}$**

Given the graph $\mathcal{H} = (V, E_{\mathcal{H}})$, we denote $\mathcal{S} = (V_{\mathcal{S}}, E_{\mathcal{S}})$ as a quotient graph through strong connectivity equivalence relation, i.e., $i \sim j \iff i$ and $j$ are strongly connected. More precisely:

*Definition* Given the graph $\mathcal{H} = (V, E_{\mathcal{H}})$, we denote $\mathcal{S} = (V_{\mathcal{S}}, E_{\mathcal{S}})$ as a reduced graph where:

- The set of vertices is the quotient set, i.e., $V_{\mathcal{S}} = V/ \sim = \{SCC(v) : v \in V\}$
- Two equivalence classes $SCC(u), SCC(v) \in V_{\mathcal{S}}$ forms an edge if and only if $(u, v) \in E_{\mathcal{H}}$. In particular (Bloem et al., 2006):

$$E_{\mathcal{S}} = \{(C, C') \,|\, C \neq C' \text{ and } \exists v \in C, v' \in C' : (v, v') \in E_{\mathcal{H}}\} \tag{7}$$

**Graph Density of Digraphs**:

Graph density computes ratio of number of edges in the graph to the maximal number of edges, i.e.,

$$d = \frac{m}{n(n-1)} \tag{8}$$

where $n$ is the number of nodes and $m$ is the number of edges in the directed graph.

## APPENDIX C   PROOF OF THE THEOREM 1 AND COROLLARY 1.1.

We restate the Theorem 1:

### C.1   THEOREM 1

For $i, j, k \in D$, assume that

$$\max_{j \in S \subseteq D} |u(S \cup \{i\}) - u(S)| \leq \varepsilon_j \qquad (I)$$

$$\max_{i \in S \subseteq D} |u(S \cup \{k\}) - u(S)| \leq \varepsilon_i \qquad (II)$$

Then, the following inequalities hold:

$$|E^2(u)_{ij}| \leq \frac{d!}{2}\varepsilon_j, |E^2(u)_{ki}| \leq \frac{d!}{2}\varepsilon_i \qquad (A)$$

$$|E^2(u)_{kj}| \leq \frac{d!}{2}(2\varepsilon_j + \varepsilon_i) \qquad (B)$$

*Proof.* **Part A)**

Using Eq (6), $|E^2(u)_{ij}|$ is equal to:

$$|E^2(u)_{ij}| = | \sum_{j \in S \subseteq D \setminus \{i\}} \frac{|S|! \, (d - |D| - 1)!}{d!}(u(S \cup \{i\}) - u(S))| \leq \tag{9}$$

$$\sum_{j \in S \subseteq D \setminus \{i\}} \frac{|S|! \, (d - |S| - 1)!}{f!} |(u(S \cup \{i\}) - u(S))| \quad \text{triangular inequality} \tag{10}$$

$$\leq \sum_{j \in S \subseteq D \setminus \{i\}} \frac{|S|! \, (d - |S| - 1)!}{d!}\varepsilon_j = ( \sum_{j \in S \subseteq D \setminus \{i\}} \frac{|S|! \, (d - |S| - 1)!}{d!})\varepsilon_j =^* \frac{d!}{2}\varepsilon_j \tag{11}$$

*:All the possible combinations of features are $d!$ but half of these times $j \in S$ and half of these times $j \notin S$, because given a sequence of features $a_1, \ldots, a_d$ where $j \in S$ as follows:

$$\underbrace{(a_1 \ldots j \ldots a_{|S|})}_{|S|} \quad i \quad \underbrace{(a_{|S|+2} \ldots a_d)}_{d-|S|-1} \tag{12}$$

There is an exact sequence on $j \notin S$ as follows:

$$\underbrace{(a_{|S|+2} \ldots a_d)}_{d-|S|-1} \quad i \quad \underbrace{(a_1 \ldots j \ldots a_{|S|})}_{|S|} \tag{13}$$

so it is $\frac{d!}{2}$ elements that $j \in S$, similarly one can derive the other inequality in part A which is $|E_{ki}^2| \leq \frac{d!}{2}\varepsilon_i$.

**Part B)**

We start by writing $E_{kj}^2$ from Eq (6), i.e.:

$$\begin{aligned}
|E_{kj}^2(u)| = | & \sum_{j \in S \subseteq D \setminus \{k\}} \frac{|S|! \, (d-|S|-1)!}{d!}(u(S \cup \{k\}) - u(S))| \\
\leq & \sum_{j \in S \subseteq D \setminus \{k\}} \frac{|S|! \, (d-|S|-1)!}{d!}|(u(S \cup \{k\}) - u(S))|
\end{aligned} \tag{14}$$

where we used triangular inequality, now we look at the element inside the summation separately when $i \in S$ and $i \notin S$, note that in all cases $j \in S$, in particular we have:

- if $i \in S$, then from the assumption 2 we have $|u(S \cup \{k\}) - u(S)|$ is less or equal than $\varepsilon_i$
- if $i \notin S$: In this case we have the following:

$$\begin{aligned}
|u(S \cup \{i\}) - u(S)| &\leq \varepsilon_j, & \text{we use (I)} \\
|u(S \cup \{i\} \cup \{k\}) - u(S \cup \{i\})| &\leq \varepsilon_i, & i \in S \cup \{i\}, \text{we use (II)} \\
|u(S \cup \{i\} \cup \{k\}) - u(S \cup \{k\})| &\leq \varepsilon_j & j \in S \cup \{k\}, \text{we use (I)}
\end{aligned} \tag{15}$$

Using these three inequalities we have:

$$|[u(S \cup \{i\}) - u(S)] + [u(S \cup \{i\} \cup \{k\}) - u(S \cup \{i\})] - [u(S \cup \{i\} \cup \{k\}) - u(S \cup \{k\})]| =$$

$$|u(S \cup \{k\}) - u(S)| \overset{**}{\leq} |[u(S \cup \{i\}) - u(S)]| + |[u(S \cup \{i\} \cup \{k\}) - u(S \cup \{i\})]| +$$

$$|[u(S \cup \{i\} \cup \{k\}) - u(S \cup \{k\})]| \leq \varepsilon_i + \varepsilon_j + \varepsilon_j \tag{16}$$

where ** uses triangular inequality. Hence we have each element is at most $2\varepsilon_j + \varepsilon_i$ for both cases when $i \in S$ or $i \notin S$, thus we have:

$$\max_{j \in S \subseteq D} |(u(S \cup \{k\}) - u(S))| \leq (2\varepsilon_j + \varepsilon_i) \tag{17}$$

using the similar arguments as in part A, we have the following:

$$\begin{aligned}
|E_{kj}^2| = & \leq \sum_{j \in S \subseteq D \setminus \{k\}} \frac{|S|! \, (d-|S|-1)!}{d!}|(u(S \cup \{k\}) - u(S))| \\
= & \sum_{j \in S \subseteq D \setminus \{k\}} \frac{|S|! \, (d-|S|-1)!}{d!}(2\varepsilon_j + \varepsilon_i) = \frac{d!}{2}(2\varepsilon_j + \varepsilon_i)
\end{aligned} \tag{18}$$

$\square$

## C.2 COROLLARY 1.1.

For the corollary we did not mention what $u$ is, to compute $E^2(u)$, we need $u$. In this corollary we assume that the utility function is monotone. In particular, Utility function $u : P(D) \to \mathbb{R}$ is monotone iff

$$\forall S, S' \text{ s.t } S \subseteq S' \subseteq P(D) \implies u(S) \leq u(S').$$

This assumption on utility states that more features given to the model does not hurt. An exmple of such utility function is mutual information, i.e., $u(S) = I(X_S; Y)$.

**Corollary 1.1.** *(Transitivity): If $E$ is the Shapley explanation map and $u$ be a monotone utility function, then graph $\mathcal{H}$ is transitive.*

*Proof.* If $E^2(u)_{ij} = 0$ and $E^2(u)_{ki} = 0$ we want to show $E^2(u)_{kj} = 0$

Based on the assumption $u$ is monotone, hence every marginal gain is greater or equal than zero, i.e., $u(S \cup \{i\}) - u(S) \geq 0$, for all $S \subseteq P(D)$ an $i \in D$.

Based on the assumption we have $E^2(u)_{ij} = 0$, i.e.,

$$E^2(u)_{ij} = 0 \implies \sum_{j \in S \subseteq D \setminus \{k\}} \frac{|S|! \, (d - |S| - 1)!}{d!} (u(S \cup \{i\}) - u(S)) = 0 \qquad (19)$$

But every element of the sum is greater or equal than zero hence, $\max_{j \in S \subseteq D} |u(S \cup \{i\}) - u(S)| = 0$, similarly from $E^2(u)_{ki} = 0$ we have $\max_{i \in S \subseteq D} |u(S \cup \{k\}) - u(S)| = 0$. Using Theorem 1 result we have:

$$|E^2_{kj}| \leq \frac{d!}{2} (2\varepsilon_j + \varepsilon_i) = 0 \implies E^2_{kj} = 0. \qquad (20)$$

$\square$

## APPENDIX D   PAGERANK

### D.1   PAGERANK

PageRank (Page et al., 1999) is an algorithm used by Google search in order to give a importance ranking for web pages in their search engine. Page rank output is a probability distribution which represent the likelihood of a person random clicking on different links to end up in a specific web page form (Page et al., 1999).In here we overview the PageRank algorithm. The PageRank scores $s_i \in [0, 1]$, where $\sum_{i \in V} s_i = 1$, are given as the solution of the following system of equations:

$$s_i = p_i \cdot \alpha + \sum_{j:(j,i) \in E} \frac{w_{ji}}{d_j} s_j \quad \text{for all } i \in V,$$

where $\alpha \in [0, 1]$ is a dampening factor (default value of 0.85),

$$d_j = \sum_{k:(j,k) \in E} w_{jk}$$

is the outgoing weighted degree of node $j \in V$ and $[p_i]_{i \in V}$ is a probability distribution over $V$. In standard PageRank, $p_i = \frac{1}{|V|}$, i.e., $p$ is the uniform distribution. In personalized pagerank, a different distribution, possibly differentiated per node, is used.

Intuitively, the PageRank scores correspond to the steady state random walk over the weighted graph with random restarts: with probability $(1 - \alpha)$ the walker transitions to an edge selected with a probability proportional to neighboring edge weights. With probability $\alpha$, the walker jumps to a random node in $V$, sampled from probability distribution $p$. Usually, they are via iterative applications of the above random walk transition equations, applied to a starting distribution over $V$ (Newman, 2018; Page et al., 1999).

## APPENDIX E   DERIVATION OF BIVARIATE SHAPLEY EXPLANATION MAP FORMULA

To prove the equation (6), we need to compute the $E^2(u)_{ij}$ elements of the matrix $E^2(u)_{ij}$. $E^2(u)_{ij}$ is the element in intersection $j^{\text{th}}$ column and $i^{\text{th}}$ row. Based on the definition of $E^2$, we $j^{\text{th}}$ column is represented as $E(u_j)$ where $u_j$ is defined as in eq (5). In the case of shapley explanation $E(u_j)_i$ has specific form based on Shapley value eq (4), i.e.,

$$E^2(u)_{ij} = E(u_j)_i = \sum_{S \subseteq D \setminus \{i\}} \frac{|S|! \, (d-|S|-1)!}{d!} (u_j(S \cup \{i\}) - u_j(S)) \tag{21}$$

From the definition of $u_j$ we know it is zero if $j \notin S$, thus we can remove those from the summation, i.e.,

$$\begin{aligned}
E^2(u)_{ij} = \sum_{j \in S \subseteq D \setminus \{i\}} \frac{|S|! \, (d-|S|-1)!}{d!} (u_j(S \cup \{i\}) - u_j(S)) + \\
\sum_{j \notin S, i \subseteq D \setminus \{i\}} \frac{|S|! \, (d-|S|-1)!}{d!} (u_j(S \cup \{i\}) - u_j(S)) = \\
\sum_{j \in S \subseteq D \setminus \{i\}} \frac{|S|! \, (d-|S|-1)!}{d!} (u(S \cup \{i\}) - u(S)) + 0
\end{aligned} \tag{22}$$

## APPENDIX F   EXTENSION OF BIVARIATE EXPLANATION MAP TO MULTIVARIATE EXPLANATION

To generalize the bivariate explanation map $E^2$, define $E^k : \mathcal{U} \to \mathbb{R}^{\overbrace{d \times \cdots \times d}^{k \text{ times}}}$, which outputs a tensor, let $T$ be the tensor output, each element of this tensor would be denoted as $T^{i_1 \cdots i_k} \in \mathbb{R}$, hence each column would be defined as $T^{i_1 \cdots i_{k-1}} \in \mathbb{R}^d$ and similar to $E^2$ is defined as $E(u_{i_1 \ldots i_{k-1}})$, where $u_{i_1 \ldots i_{k-1}}$ is defined as follows:

$$u_{i_1 \ldots i_{k-1}} : P(D) \to \mathbb{R} \implies \forall S \in P(D), \, u_{i_1 \cdots _{k-1}}(S) = \begin{cases} u, \text{ if } \{i_1, \ldots i_{k-1}\} \subseteq S \\ 0, \text{ if } \{i_1, \ldots i_{k-1}\} \not\subseteq S \end{cases} \tag{23}$$

## APPENDIX G   DETAILS OF EXPERIMENTAL SETUP AND ADDITIONAL EXPERIMENTAL RESULTS

### G.1   EXPERIMENT SETUP

#### G.1.1   ALGORITHMS

**Approximating Graph $\mathcal{G}$ with Shapley Sampling.** Computation over all subsets of features is computationally expensive. In practice we can use a approximate the Bivarate Shapley value over a fixed number of samples by adapting the sampling algorithm introduced by Štrumbelj & Kononenko (Štrumbelj & Kononenko, 2014), as seen in Alg. 1. Note that computing the $\mathcal{G}$ matrix adds no complexity to the original algorithm; we simply keep track of when feature $j$ is absent (i.e. $j \notin S$ and set the value function output to zero when this condition occurs. We can therefore calculate the bivariate and univariate shapley values concurrently. Note that since we are discarding or "filtering out" the samples where $j$ is absent, we need to double the number of samples to achieve the same approximation accuracy as the univariate calculation.

**Approximating Graph $\mathcal{G}$ with KernelSHAP.**

As mentioned in Section 4, our method can be generalized to any removal-based Shapley approximation or other removal-based explainer. More concretely, each column of the $d \times d$ interaction matrix, representing interactions between d features, can be considered an independent explanation where the column feature is always present. Therefore our method is extremely flexible; the user can decide which explanation method to use based on the constraints of their intended application. However, we can also improve performance by taking advantage of different approximation methods.

---

**Algorithm 1** Approximate Graph $\mathcal{G}$ with Shapley Sampling Algorithm

---

**Input :** Data Sample $x \in \mathcal{X} \subset \mathbb{R}^d$, Utility Function $f$, Number of Samples $M$
**Output :** Adjacency Matrix $\mathcal{G} \in \mathbb{R}^{d \times d}$

---

Initialize $\mathcal{G} = 0$
**for** $i=1...d$ **do**
   **for** $m=1...M$ **do**
      Create random permutation $\mathcal{O}$ of size $d$
      Define $\tilde{i}$ as permuted index of feature $i$
      Define the set of indices $s = \{\mathcal{O}_{1...\tilde{i}-1}\}$ and the set of all indices $D = \{\mathcal{O}\}$
      Sample random baseline $w \in \mathcal{X}$

      $b_1 = x_{s \cup i} \oplus w_{D \setminus \{s \cup i\}}$         $\setminus \setminus$ Symbol $\oplus$ indicates concatenation
      $b_2 = x_s \oplus w_{D \setminus s}$

      **for** $j \in s$ **do**
         $\mathcal{G}_{ij} = \mathcal{G}_{ij} + f(b_1) - f(b_2)$
      **end**
   **end**
**end**
$\mathcal{G} = \frac{1}{M} \mathcal{G}$

Return $\mathcal{G}$

---

Performance is improved through the use of two properties in KernelSHAP. First, we can save the sampled model outputs and reuse these values when recalculating KernelSHAP over all $d$ features. This allows for a fixed number of model samples independent of the number of data features. Second, note that KernelSHAP attributions are calculated through a weighted linear regression mechanism. Bivariate Shapley simply changes the model output (setting the output to zero) depending on whether a feature is present or removed, which corresponds to applying a binary mask over the linear regression labels. Therefore we can save the intermediate linear regression calculation and evaluate each column of the interaction matrix with two matrix multiplications with the sparse labels. This results in significant speed improvements and allows scaling to datasets with large number of features, such as the COPD dataset with 1,077 features.

**Mutual Redundancy on Graph $\mathcal{H}$.** Given the unweighted graph $\mathcal{H}$, we want to find groups of mutually redundant features as investigated in Fig. 2. These features are identified as strongly connected nodes within the graph. We use the package NetworkX (Schult, 2008), which implements Tarjan's algorithm (Tarjan, 1972) to identify such nodes. Tarjan's algorithm is a depth-first search that runs in linear time. This algorithm for identifying mutually redundant features is outlined in Alg. 3.

To generate the results of Fig. 2, we apply a binary mask for each sample such that a given percentage of its mutually redundant features are set to their baseline value. Note that the number of features masked for a given percentage may vary between samples, since the $\mathcal{H}$ is calculated on an instance-wise basis. We then record the accuracy for the set of masked samples.

**Directional Redundancy on Graph $\mathcal{H}$.** Directional redundancy is defined in terms of $\mathcal{H}$-sinks and $\mathcal{H}$-sources on graph $\mathcal{H}$, which we investigate in Table 2. There are a number of methods to identify source and sink nodes on a graph; in our implementation we use the PageRank algorithm (Page et al., 1999) and take the maximally and minimally-ranked node as the sink and source, respectively (Alg. 4). Note that using PageRank in such manner will only identify singular sinks and sources, therefore we first separate graph $\mathcal{H}$ into its connected subgraphs and apply PageRank to the condensation graph of each subgraph. We use the PageRank implementation in the Scikit-Network package (Bonald

---

[5]Note that $\pi(x) = \infty$ when $|x| \in \{0, d\}$, therefore in practice we remove two variables during the linear regression calculation and enforce the following two constraints. 1) $\phi_0 = f(\mathbb{E}[\mathcal{X}])$, where $\phi_0$ is defined as the bias / intercept term in the regression, and 2) $\sum_i \phi_i = f(x) - f(\mathbb{E}[\mathcal{X}])$.

---

**Algorithm 2** Approximate Graph $\mathcal{G}$ with KernelSHAP Algorithm

---

**Input :** Data Sample $x \in \mathcal{X} \subset \mathbb{R}^d$, Utility Function $f$, Number of Samples $M$
**Output :** Univariate Shapley Values $\phi \in \mathbb{R}^d$, Adjacency Matrix $\mathcal{G} \in \mathbb{R}^{d \times d}$

Initialize $\mathcal{G} \in \mathbb{R}^{d \times d}$
Initialize matrix $\tilde{X} \in \{0,1\}^{M \times d}$
Initialize matrix $\Pi$ as an identity matrix $\mathbb{I}_M$
Initialize vector $Y \in \mathbb{R}^M$
Define the KernelSHAP weighting kernel $\pi(x) = \frac{(d-1)}{(d \text{ choose } |x|)|x|(d-|x|)}$

\\ Randomly draw $M$ samples around $x$
**for** *m=1...M* **do**
  Sample random baseline $w \in \mathcal{X}$
  Sample binary vector $\tilde{x} \in \{0,1\}^d$
  Calculate perturbed labels $y = f(\tilde{x} \odot x + (1 - \tilde{x}) \odot w)$        \\ Symbol $\odot$ indicates Hadamard product

  $\tilde{X}_{m,:} \leftarrow \tilde{x}$            \\$X_{i,j}$ indicates indices i and j in matrix X. : indicates the entire row / column.
  $Y_m \leftarrow \tilde{y}$
  $\Pi_{m,m} \leftarrow \pi(\tilde{x}_m)$
**end**

\\ Solve a constrained [5], weighted linear regression
Define $\Gamma = (\tilde{X}^T \Pi \tilde{X})^{-1} \tilde{X}^T \Pi$
Define $\Gamma^+ = \Gamma_{-d,:}$            \\ remove the last row of $\Gamma$
Define $\Gamma^- = \Gamma_{-1,:}$            \\ remove the first row of $\Gamma$

\\ Remove the last feature in regression calculation with the constraint that $\sum_i \phi_i = f(x) - f(\mathbb{E}[\mathcal{X}])$
Define $\phi = \Gamma^+ [Y - X_{:,d} \times (f(x) - f(\mathbb{E}[\mathcal{X}]))]$
$\phi \leftarrow \phi \oplus (f(x) - f(\mathbb{E}[\mathcal{X}]) - \sum_i \phi_i)$            \\ Enforce constraint. $\oplus$ indicates concatenation.

\\ Iterate regression calculation over filtered labels
**for** *j=1...d* **do**
  Define $Y^+ = Y \odot \tilde{X}_{:,j}$            \\ Set $Y_m = 0$ if feature $j$ was not selected in $\tilde{X}_{m,:}$
  Define $\phi^+ = \Gamma^+ [Y^+ - X_{:,d} \times f(x)]$

  Define $Y^- = Y \odot (1 - \tilde{X}_{:,j})$            \\ Set $Y_m = 0$ if feature $j$ was selected in $\tilde{X}_{m,:}$
  Define $\phi^- = \Gamma^- [Y^- + X_{:,d} \times f(\mathbb{E}[\mathcal{X}])]$

  $\phi^+ \leftarrow \phi^+ \oplus (\phi_d - \phi^-_{d-1})$            \\ Utilize the property that $\phi = \phi^- + \phi^+$
  $\mathcal{G}_{:,j} \leftarrow \phi^+$
**end**

Return $\phi, \mathcal{G}$

---

et al., 2020) with no personalization and default damping $= 0.85$. In practice, we found that changing the damping parameter had no effect on identified features.

In Table 2 we show the results of completely masking all $\mathcal{H}$-sources and $\mathcal{H}$-sinks. We apply a binary mask, setting the value of all $\mathcal{H}$-source or $\mathcal{H}$-sink features to their baseline values, then record the sample accuracy.

**Redundancy Ranking on Graph $\mathcal{G}$.** We want to create a continuous ranking of feature redundancy given graph $\mathcal{G}$, as investigated in Fig. 3. We first add $\epsilon = 10^{-70}$ to each element in $\mathcal{G}$ to eliminate disconnected subgraphs. Note that for certain value functions, such as those used in our experiments, the graph $G$ can contain negative values. We normalize these negative values by applying an element-

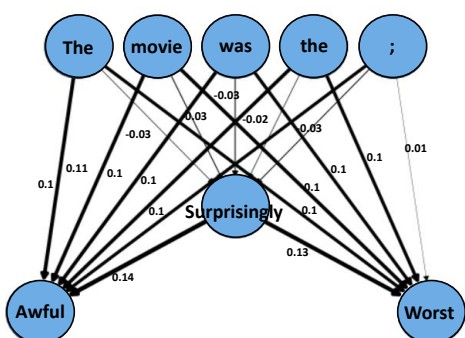 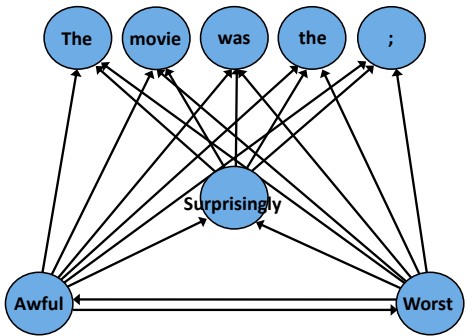

Figure 5: Comparison of graph $\mathcal{G}$ and graph $\mathcal{H}$ for the the given IMDB example "The movie was the worst; surprisingly awful", which is classified as negative sentiment. Note that the sinks and sources of graph $\mathcal{G}$ and graph $\mathcal{H}$ are reversed in terms of influential and redundant features. I.E. the **source** nodes of graph $\mathcal{G}$ represent redundant features, whereas the **sink** nodes of graph $\mathcal{H}$ represent (directionally) redundant features.

---

**Algorithm 3** Mutual Redundancy on Graph $\mathcal{H}$

---

**Input :** Unweighted Directed Graph $\mathcal{H}$
**Output :** Groups of Mutually Redundant Features

Define $\mathcal{S} = \{s_1, ..., s_m\}$ as the set of strongly connected subgraphs in $\mathcal{H}$

Return $\mathcal{S}$

---

**Algorithm 4** Directional Redundancy on Graph $\mathcal{H}$

---

**Input :** Unweighted Directed Graph $\mathcal{H}$
**Output :** Source Nodes and Sink Nodes

Define $\mathcal{W} = \{w_1, ..., w_m\}$ as the set of weakly connected subgraphs in $\mathcal{H}$
**for** $i = 1,...,m$ **do**
    Create condensed graph $c_i = \text{Condensation}(w_i)$
    Source Node $\alpha_i = \text{argmax PageRank }(c_i)$
    Sink Node $\omega_i = \text{argmin PageRank }(c_i)$

**end**
Source Nodes = $\text{Condensation}^{-1}(\{\alpha_1, ..., \alpha_m\})$
Sink Nodes = $\text{Condensation}^{-1}(\{\omega_1, ..., \omega_m\})$

Return Source Nodes, Sink Nodes

---

wise Softplus function: $\text{Softplus}(x) = \ln(1 + e^x)$. We then directly apply the PageRank algorithm from Scikit-Network to obtain feature rankings. We again use the default damping parameter of $0.85$ for all datasets.

One issue we observed during testing was the occurrence of nodes with identical PageRank scores, indicating a similar level of redundancy. With no other information, this would necessitate random selection when generating the feature ranking. With this motivation, we experiment with using the univariate shapley values as personalization values. In personalized PageRank, the personalization

values dictate the distribution over nodes for which a random jump will land. With no personalization, a random jump will land in each node with equal probability; i.e. the personalization is assumed to be uniform. By setting the personalization to the univariate shapley values, we bias the stationary distribution towards nodes that have high shapley values. Therefore, nodes of similar redundancy would be further ranked by their respective univariate shapley values. In practice, using personalization slightly improves post-hoc accuracy results in Fig. 3 for larger masking percentages. The full algorithm for generating the redundancy ranking of features is outlined in Alg. 5.

---

**Algorithm 5** Directional Redundancy Ranking on Graph $\mathcal{G}$.

---

**Input :** Weighted Directed Graph $\mathcal{G}$, *optional* Univariate Feature Ranking $R \in \mathbb{R}^d$
**Output :** Score vector $S \in \mathbb{R}^d$, representing relative feature importance for each feature.

Define $A$ as the adjacency matrix for Graph $\mathcal{G}$

$\tilde{A} = A + 10^{-70}$          \\ Add $\epsilon \approx 0$ to ensure all nodes are connected
$\tilde{A} = \text{SoftPlus}(A)$       \\ Element-wise Softplus function to normalize negative values

**if** *Personalization* **then**
   |   $S = \text{PageRank}(\tilde{A})$ with Personalization Values $R$
**else**
   |   $S = \text{PageRank}(\tilde{A})$

**end**
Return $S$

---

### G.1.2   IMPLEMENTATION DETAILS FOR BIVARIATE SHAPLEY AND COMPETITORS.

Unless otherwise specified, we use the default parameters when implementing comparison methods using publicly available code.

Removal-based methods typically assign a value to act as a proxy for a feature's absence during feature removal. This value is commonly referred to as a baseline, or reference value, and is often assigned to be some *a priori* neutral value. While different removal-based methods may have different baseline values as default, we assign a single baseline value used for all methods for a given dataset. This is to maintain comparability, since the objective of our experiments is to evaluate the explanation calculation rather than the choice of baseline value. For tabular data, we define the value for all removed features to be zero, except the Divorce dataset where a value of '3' indicates the average response, and the Census dataset where we fix the baseline to be the average value for each feature. For images, we use a pixel value of zero. For text, we set the word embedding for the selected feature to be the zero vector.

**Bivariate Shapley - Sampling.** We apply Bivariate Shapley on a variety of prediction models (detailed in section G.1.3), using a value function $v(S) = \mathbb{E}_{w \sim \mathcal{B}}[P(Y = \hat{y} | X = x_S \cup w_{\bar{S}})]$, where $\hat{y}$ is the model's predicted class, $\bar{S}$ is the complement of $S$, and $w$ represents samples drawn from a baseline distribution $\mathcal{B}$. As previously discussed, this baseline distribution is fixed to a value that is dependent on the given dataset. We set $m$, the number of samples drawn in alg. 1 to be 1000.

**Bivariate Shapley - Kernel.** We utilize the algorithm described in sec.G.1.1 by adapting the publicly available package for kernelSHAP (Lundberg & Lee, 2017). We keep the same default parameters as KernelSHAP, except we double the number of default samples to account for the Bivariate Shapley filtering.

**Shapley Excess.** Shapley Excess refers to the surplus value from contribution of players in a coalition game grouped in a singleton coalition as compared to their individual contributions. This can be written formally as:

$$\phi_s - \sum_{i \in s} \phi_i$$

where $\phi_s$ is the shapley value of a group of players when considered as a singleton player. We implement this formula using the KernelSHAP approximation by combining features and evaluating the resulting excess Shapley value.

**Shapley Interaction Index.** Introduced by (Grabisch & Roubens, 1999), Shapley Interaction Index has gained popularity due to the efficient implementation by (Lundberg et al., 2018b) on tree-based prediction models. In order to apply this method efficiently with the entirety of the datasets in our experiments, we use the KernelSHAP approximation to calculate Shapley Interaction Index. This implementation results in significantly faster calculations compared to Shapley Sampling approximations (as seen in tbl 8. We use the default parameters of KernelSHAP, applied $2 \times d$ times per sample, where $d$ is the number of features.

**Shapley Excess.** Shapley Excess refers to the surplus value from contribution of players in a coalition game grouped in a singleton coalition as compared to their individual contributions. This can be written formally as:

$$\phi_s - \sum_{i \in s} \phi_i$$

where $\phi_s$ is the shapley value of a group of players when considered as a singleton player. We implement this formula using the KernelSHAP approximation by combining features and evaluating the resulting excess Shapley value.

**Shapley Taylor Index.** Introduced by Sundararajan et al. (2020b). As of this writing, there is no publicly available code for the Shapley Taylor Index. Therefore we build our own implementation using the Shapley Sampling approximation as outlined in the original paper. We choose a sample size of $m = 200$ for each element of the interaction matrix.

**GNNExplainer.** Introduced by Ying et al. (2019), GNNExplainer is a method for explaining a GNN-based black-box model. It can be used on a variety of GNN tasks, such as node or graph classification, and identifies a compact subgraph and subset of node features that best explains the GNN output. This is accomplished through the use of a soft mask on the edges and node features of the input graph. Specifically, GNNExplainer trains a neural network to generate the edge and node feature masks, with the objective of maximizing mutual information between the black-box output of the masked graph and the label.

While GNNExplainer was originally intended to explain GNN models, it can be used in conjunction with non-GNN models. In our implementation, our objective is to identify the important edges between the features of a data sample. Therefore we define a fully-connected graph with the features as the graph nodes. When applying GNNExplainer to this fully-connected graph, GNNExplainer returns edge importance values for the given data sample. This output can be converted to a subgraph using specifying a threshold, below which the edges are removed. In our experiments, we directly use the edge importance values as the weights of a directed, weighted graph. This resulting graph is then evaluated and compared with Bivariate Shapley using the same algorithms for identifying mutually redundant features, directionally redundant features, and feature redundancy ranking, as outlined in App. G.1.1. We implement GNNExplainer using the Pytorch Geometric package (Fey & Lenssen, 2019) with default parameters.

Note that while GNNExplainer can indeed be applied to non-GNN models, these models may not be able to incorporate the graph structure in its predictions. For example, even though GNNExplainer applies an edge mask to the input graph, this edge information is meaningless if the black-box model is not designed to use this structure in its prediction. In this case, the GNNExplainer will receive non-informative black-box outputs in its mutual information maximization objective.

### G.1.3 DATASETS AND MODELS

**COPDGene.** The COPDGene dataset is an observational study with a cohort of 10,000 participants designed to identify the genetic risk factors for COPD. The study contains participants with and without COPD; COPD diagnosis, subtyping, and progression are monitored using high-resolution CT scans. We are interested in investing the relation between gene expression and smoking status (see section G.2.6 for details). The dataset contains RNA-sequencing count data for 1,077 genes and the associated binary label for smoking status. We use a neural network with 4 fully-connected

| Domain | Genetics | Image | | Text | Tabular | | |
|---|---|---|---|---|---|---|---|
| **Dataset** | COPDGene | CIFAR10 | MNIST | IMDB | Census | Divorce | Drug |
| **Classes** | 2 | 10 | 10 | 2 | 2 | 2 | 2 |
| **Train/Test Samples** | 1,641/407 | 50k/10k | 60k/10k | 25k/25k | 26k/6.5k | 102/68 | 1413/472 |
| **Model** | 4-Layer MLP | Resnet18 | 2-Layer CNN | 1-Layer GRU | XGBoost | 3-Layer MLP | Random Forest |
| **Model Accuracy** | 88.2 | 89.8 | 99.0 | 88.1 | 87.3 | 98.5 | 85.3 |

Table 3: Summary of the datasets and models in our investigation

layers of 200 hidden units, batch normalization, and relu activation. The model is trained using Adam (Kingma & Ba, 2017) with learning rate $10^{-3}$ for 800 epochs, achieving a test accuracy of $88.2\%$.

**CIFAR10.** CIFAR10 (Krizhevsky, 2009) consists of 60k images of dimension $32 \times 32$ with RGB channels. We train a Convolution Neural Network (CNN) to classify the 10 different classes, using a Resnet18 architecture (He et al., 2016) with default parameters. We apply color jittering and horizontal flip data augmentations, as well as data normalization. The model is trained using Adam with learning rate $10^{-3}$ for 80 epochs, achieving a test accuracy of $89.8\%$.

While it is possible to use individual pixels when calculating Bivariate Shapley, we choose to use superpixels to reduce computation and improve the interpretability of results. Superpixels are contiguous clusters of pixels that are treated as a single feature for feature importance purposes; i.e. all individual pixels within the superpixel are masked or selected jointly. We use the *simple linear iterative clustering* (SLIC) algorithm (Achanta et al., 2012) in our image experiments. SLIC divides the image into similarly sized superpixels based on clustering in the CIELAB color space. For CIFAR10, we use SLIC with 255 superpixels and minimal smoothing ($\sigma = 5$).

**MNIST.** MNIST (LeCun & Cortes, 2010) consists of $28 \times 28$ greyscale images with the handwritten numerals $0 - 9$. We train a CNN with two convolution layers and a single batch normalization layer. Each convolution uses a $6 \times 6$ kernel size, stride 2, and a 200 channel mapping. We train the model using stochastic gradient descent (SGD) with learning rate $10^{-2}$ for 20 epochs, achieving a test accuracy of $99.0\%$. We again use SLIC to create superpixels; for MNIST we use 196 superpixels and $\sigma = 5$.

**IMDB.** The Large Movie Review Dataset (IMDB) (Maas et al., 2011) consists of 50k movie reviews which we use for the task of sentiment analysis. We train a Recurrent Neural Network (RNN) classifier with a single Gated Recurrent Unit (GRU) (Cho et al., 2014) layer of 500 hidden units to predict either positive or negative sentiment. We tokenize each review using the NLTK package (Loper & Bird, 2002) and map each token to a pretrained word embedding. We use the 300-dimensional GloVe (Pennington et al., 2014) embedding with 840B tokens, pretrained on the Common Crawl dataset. We limit the vocabulary to 10k tokens and text sample length to 400 tokens. The model was trained using Adam with learning rate $10^{-4}$ for 15 epochs, achieving test accuracy of $88.1\%$.

**Census.** The UCI Census Income dataset aggregates data from the 1994 census dataset. We use 12 features, including both continuous and discrete data, to predict whether an individual has an annual income greater than $\$50k$. Our model is trained using XGBoost (Chen & Guestrin, 2016) with a maximum of 5000 trees, $\eta = 0.01$, and subsample $= 0.5$, achieving $87.3\%$ test accuracy.

**Divorce.** The UCI Divorce Predictors dataset (Yöntem et al., 2019) consists of a 54-question survey with 170 participants regarding various activities and attitudes towards their partners. Each question is answered with a ranking on a scale from $1 - 5$. We train a 3-layer MLP with relu activation, predicting if the participant was divorced. Each hidden layer contained 50 hidden units. The model was trained using SGD with learning rate 0.1 and achieved test accuracy of $98.5\%$. During the Bivariate Shapley calculation, we use a baseline value of 3 to indicate a feature's absence, as this represents the value representing a neutral response.

**Drug.** The UCI Drug Consumption dataset (Fehrman et al., 2017) consists of 1,885 responses to an online survey concerning the consumption habits of various drugs. We use binary features for the six drugs nicotine, marijuana, cocaine, crack, ecstasy, and mushrooms, indicating whether the respective drug has been previously consumed. We build a model to predict whether the participant has also consumed a seventh drug, LSD. We use a random forest model with 100 trees, achieving a test accuracy of $85.3\%$

| Clinical Center | Institution Title | Protocol Number |
|---|---|---|
| National Jewish Health | National Jewish IRB | HS-1883a |
| Brigham and Women's Hospital | Partners Human Research Committee | 2007-P-000554/2; BWH |
| Baylor College of Medicine | Institutional Review Board for Baylor College of Medicine and Affiliated Hospitals | H-22209 |
| Michael E. DeBakey VAMC | Institutional Review Board for Baylor College of Medicine and Affiliated Hospitals | H-22202 |
| Columbia University Medical Center | Columbia University Medical Center IRB | IRB-AAAC9324 |
| Duke University Medical Center | The Duke University Health System Institutional Review Board for Clinical Investigations (DUHS IRB) | Pro00004464 |
| Johns Hopkins University | Johns Hopkins Medicine Institutional Review Boards (JHM IRB) | $NA_0 0011524$ |
| Los Angeles Biomedical Research Institute | The John F. Wolf, MD Human Subjects Committee of Harbor-UCLA Medical Center | 12756-01 |
| Morehouse School of Medicine | Morehouse School of Medicine Institutional Review Board | 07-1029 |
| Temple University | Temple University Office for Human Subjects Protections Institutional Review Board | 11369 |
| University of Alabama at Birmingham | The University of Alabama at Birmingham Institutional Review Board for Human Use | FO70712014 |
| University of California, San Diego | University of California, San Diego Human Research Protections Program | 70876 |
| University of Iowa | The University of Iowa Human Subjects Office | 200710717 |
| Ann Arbor VA | VA Ann Arbor Healthcare System IRB | PCC 2008-110732 |
| University of Minnesota | University of Minnesota Research Subjects' Protection Programs (RSPP) | 0801M24949 |
| University of Pittsburgh | University of Pittsburgh Institutional Review Board | PRO07120059 |
| University of Texas Health Sciences Center at San Antonio | UT Health Science Center San Antonio Institutional Review Board | HSC20070644H |
| Health Partners Research Foundation | Health Partners Research Foundation Institutional Review Board | 07-127 |
| University of Michigan | Medical School Institutional Review Board (IRBMED) | HUM00014973 |
| Minneapolis VA Medical Center | Minneapolis VAMC IRB | 4128-A |
| Fallon Clinic | Institutional Review Board/Research Review Committee Saint Vincent Hospital – Fallon Clinic – Fallon Community Health Plan | 1143 |

Table 4: IRB Information for COPDGene Dataset

### G.1.4 LICENSES FOR COPDGENE DATA

All participants provided their informed consent, and IRB approval was obtained from all concerned institutions. IRB information is provided in Tab. 4.

## G.2 ADDITIONAL EXPERIMENTAL RESULTS

### G.2.1 INSERTION AND DELETION AUC

Insertion AUC (iAUC) and Deletion AUC (dAUC), introduced by (Petsiuk et al., 2018), quantify the ability for an explainer to find the most influential features of a given black-box model. We use iAUC and dAUC as a supplementary metric to evaluate the redundancy-based ranking we explore in figure 3.

To summarize, dAUC iteratively removes the highest-ranked features of a given image and measures the change in model output compared to the baseline prediction, as summarized by the area under

| Dataset | Insertion AUC (Higher is better) | | | | | | | Deletion AUC (Lower is better) | | | | | | |
|---|---|---|---|---|---|---|---|---|---|---|---|---|---|---|
| | COPD | CIFAR10 | MNIST | IMDB | Census | Divorce | Drug | COPD | CIFAR10 | MNIST | IMDB | Census | Divorce | Drug |
| Ours-SS | 0.48 | 0.75 | 0.85 | 0.45 | 0.43 | 0.30 | 0.30 | 0.01 | 0.05 | 0.03 | 0.02 | 0.32 | 0.05 | 0.10 |
| Ours-K | 0.49 | 0.65 | 0.84 | 0.43 | 0.42 | 0.30 | 0.30 | 0.00 | 0.08 | 0.03 | 0.02 | 0.32 | 0.05 | 0.10 |
| Sh-Sam | 0.48 | 0.75 | 0.85 | 0.45 | 0.43 | 0.30 | 0.30 | 0.01 | 0.05 | 0.03 | 0.02 | 0.32 | 0.05 | 0.10 |
| kSHAP | 0.42 | 0.48 | 0.77 | 0.29 | 0.42 | 0.29 | 0.30 | 0.09 | 0.17 | 0.17 | 0.03 | 0.36 | 0.05 | 0.11 |
| Sh-Int | 0.20 | 0.35 | 0.46 | 0.32 | 0.43 | 0.16 | 0.17 | 0.23 | 0.31 | 0.52 | 0.29 | 0.33 | 0.14 | 0.20 |
| Sh-Tay | – | 0.34 | 0.78 | 0.34 | 0.42 | 0.30 | 0.16 | – | 0.27 | 0.19 | 0.19 | 0.31 | 0.05 | 0.19 |
| Sh-Exc | – | 0.32 | 0.51 | 0.30 | 0.37 | 0.15 | 0.08 | – | 0.31 | 0.48 | 0.29 | 0.38 | 0.15 | 0.29 |
| GNNExp | 0.25 | 0.15 | 0.25 | 0.30 | 0.38 | 0.25 | 0.26 | 0.25 | 0.15 | 0.25 | 0.30 | 0.38 | 0.25 | 0.27 |

Table 5: Influential Feature Evaluation through Insertion and Deletion AUC. We calculate a feature ranking by applying PageRank on the $\mathcal{G}$ graph, iteratively removing the most influential feature, then evaluating AUC on the resulting curve. Note that we cannot run Sh-Tay and Sh-Exc methods on the COPD dataset due to their computational issues with the large number of features.

| Dataset | 10% Features Mask | | | | | 50% Features Mask | | | | |
|---|---|---|---|---|---|---|---|---|---|---|
| | Ours | Shap Sampl | Int | KernelSHAP | L2X | Ours | Shap Sampl | Int | KernelSHAP | L2X |
| COPD | $100 \pm 0.0$ | $100 \pm 0.0$ | $82.8 \pm 1.9$ | $99.3 \pm 0.4$ | $92.6 \pm 1.3$ | $100 \pm 0.0$ | $100 \pm 0.0$ | $68.3 \pm 2.3$ | $100 \pm 0.0$ | $86.0 \pm 1.7$ |
| CIFAR10 | $99.4 \pm 0.3$ | $99.0 \pm 0.4$ | $70.2 \pm 2.0$ | $86.6 \pm 1.0$ | $71.4 \pm 2.0$ | $93.0 \pm 1.1$ | $92.4 \pm 1.2$ | $32.8 \pm 2.1$ | $54.9 \pm 1.4$ | $23.2 \pm 1.9$ |
| MNIST | $100 \pm 0.0$ | $100 \pm 0.0$ | $84.6 \pm 1.6$ | $100 \pm 0.0$ | $100 \pm 0.0$ | $100 \pm 0.0$ | $100 \pm 0.0$ | $62.8 \pm 2.2$ | $99.9 \pm 0.4$ | $100 \pm 0.0$ |
| IMDB | $100 \pm 0.0$ | $100 \pm 0.0$ | $92.6 \pm 1.2$ | $100 \pm 0.0$ | $94.0 \pm 1.2$ | $100 \pm 0.0$ | $100 \pm 0.0$ | $64.4 \pm 2.1$ | $100 \pm 0.0$ | $57.9 \pm 2.2$ |
| Census | $100 \pm 0.0$ | $100 \pm 0.0$ | $100 \pm 0.0$ | $96.0 \pm 0.9$ | $96.6 \pm 0.8$ | $96.8 \pm 0.8$ | $96.8 \pm 0.8$ | $94.8 \pm 1.0$ | $90.0 \pm 1.3$ | $84.8 \pm 1.6$ |
| Divorce | $100 \pm 0.0$ | $100 \pm 0.0$ | $98.5 \pm 1.5$ | $100 \pm 0.0$ | $100 \pm 0.0$ | $100 \pm 0.0$ | $100 \pm 0.0$ | $58.8 \pm 6.0$ | $98.5 \pm 1.5$ | $98.5 \pm 1.5$ |
| Drug | $100 \pm 0.0$ | $100 \pm 0.0$ | $91.7 \pm 1.3$ | $100 \pm 0.0$ | $100 \pm 0.0$ | $99.2 \pm 0.4$ | $99.2 \pm 0.4$ | $77.1 \pm 1.9$ | $100 \pm 0.0$ | $75 \pm 2.0$ |

Table 6: Accuracy results for masking redundant features as identified using PageRank on graph $\mathcal{G}$. These results mirror Figure 3 but with the result variance, as represented by $\pm$ standard deviation. Note that for datasets with $< 10$ features, the given feature mask percentage is approximate.

the curve. Lower dAUC values indicate that the explainer can accurately assess the features most influential towards the model output. Conversely, iAUC starts with an uninformative baseline sample then iteratively inserts the highest-ranked features, then measures change in model output through AUC. Higher values of iAUC indicate better performance. We evaluate Bivariate Shapley, as well as a variety of popular univariate and bivariate black-box explainers, on these two metrics in table 5.

## G.2.2 SAMPLING VARIANCE OF POST-HOC ACCURACY RESULTS

The Bivariate Shapley method, like other shapley-based methods, does not involve any training or optimization of weights. Therefore it does not suffer from issues related to data variability. In addition, the quantitative results from our experiments are averaged over $\approx 500$ test samples (less for divorce and drug, due to dataset size). We show the variance results for Figure 3 in Table 6.

## G.2.3 BIVSHAP-K RESULTS FOR SINK AND SOURCE MASKING ON GRAPH $\mathcal{H}$

The BivShap-K results for sink and source masking on graph $\mathcal{H}$ are omitted in table 2 due to space constraints. We present the full full results in table 7.

## G.2.4 SENSITIVITY OF GRAPH $\mathcal{H}_\gamma$ TO $\gamma$

In Section 4.1 we define a relaxed version of the redundancy graph $\mathcal{H}_\gamma = (V_\mathcal{H}, E_\mathcal{H}^\gamma)$ where $V_\mathcal{H} = V_\mathcal{G}$ and $E_\mathcal{H}^\gamma = \{(i,j) \in E_\mathcal{G} : |W_\mathcal{G}(i,j)| \leq \gamma\}$. Intuitively, $\gamma \in \mathcal{R}^+$ acts as a threshold to define redundant edges in $\mathcal{H}_\gamma$. As $\gamma$ increases, the number of edges in $\mathcal{H}_\gamma$ also increases, resulting in larger mutually redundant clusters and a higher sensitivity to directional redundancy. From the perspective of accurately representing the black-box model, the choice of $\gamma$ presents a tradeoff akin to sensitivity and specificity: larger $\gamma$ values more easily identify true redundancies within the data (increased sensitivity), at the cost of potentially mislabeling non-redundancies (reduced specificity).

While it is trivial to choose $\gamma$ through cross-validation using post-hoc accuracy (or equivalent metric), such methods are not ideal for instance-wise explanation purposes where the practitioner may not have access to a sufficient number of validation samples. We therefore attempt to establish guidelines for choosing $\gamma$. In particular we investigate the effect of $\gamma$ on graph density (Figure 6). Note that

| Bivariate Shapley-S | | | | |
|---|---|---|---|---|
| | PH-Accy | | % Feat Masked | |
| Dataset | Sink Masked | Source Masked | Sink Masked | Source Masked |
| COPD | 99.5 | 62.7 | 1.5 | 98.5 |
| CIFAR10 | 94.6 | 15.0 | 6.2 | 93.8 |
| MNIST | 100.0 | 13.4 | 77.7 | 22.3 |
| IMDB | 100.0 | 54.0 | 3.5 | 96.5 |
| Census | 100.0 | 82.0 | 23.8 | 76.2 |
| Divorce | 100.0 | 51.5 | 22.2 | 77.8 |
| Drug | 100.0 | 48.5 | 43.5 | 56.5 |

| Bivariate Shapley-K | | | | |
|---|---|---|---|---|
| | PH-Accy | | % Feat Masked | |
| Dataset | Sink Masked | Source Masked | Sink Masked | Source Masked |
| COPD | 97.3 | 62.7 | 13.6 | 86.4 |
| CIFAR10 | 82.6 | 19.4 | 10.4 | 89.6 |
| MNIST | 100.0 | 17.6 | 13.6 | 86.4 |
| IMDB | 97.2 | 54.0 | 23.7 | 76.3 |
| Census | 100.0 | 82.0 | 33.3 | 66.7 |
| Divorce | 100.0 | 51.5 | 22.0 | 78.0 |
| Drug | 100.0 | 48.5 | 43.5 | 56.5 |

Table 7: Posthoc-accy of BivShap-S and BivShap-K after masking $\mathcal{H}$-source nodes, representing features with minimal redundancies, and $\mathcal{H}$-sink nodes, representing directionally redundant features.

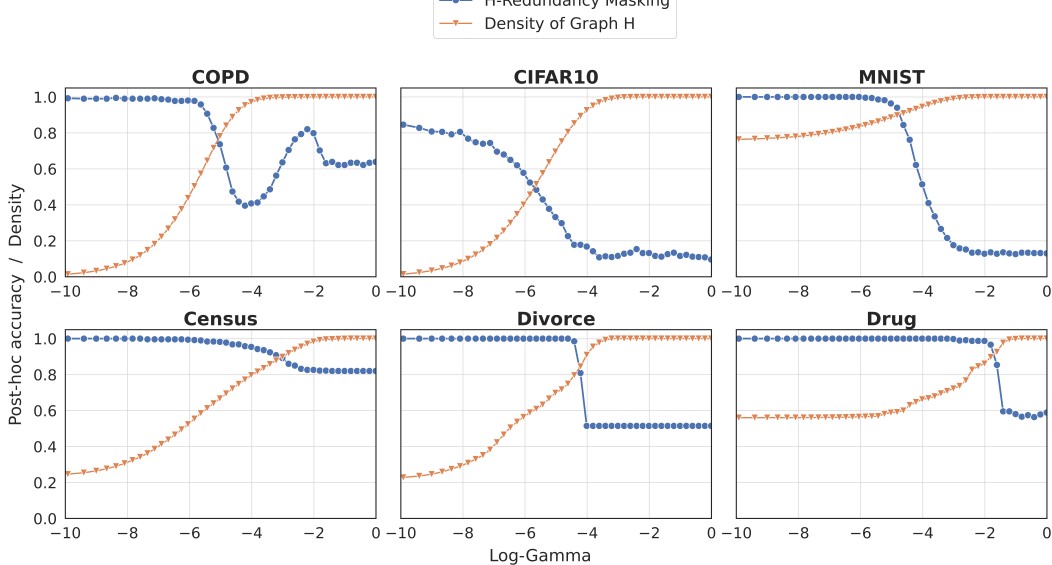

Figure 6: Sensitivity analysis of graph $\mathcal{H}$ to parameter $\gamma$. We compare the density of $\mathcal{H}$ as $\gamma$ increases to the post-hoc accuracy after masking all directionally redundant features found in $\mathcal{H}$.

increasing $\gamma$ also increases the density of graph $\mathcal{H}$. We can see that at a certain density, post-hoc accuracy exhibits a sharp decrease, suggesting that the identified redundancies are not truly redundant. The ideal $\gamma$ depends on the level of redundancy in the dataset, therefore the value should be chosen based on the given task. For experimental purposes, we use a constant $\gamma = 10^{-5}$ for all datasets.

We also explore how increasing $\gamma$ affects the identification of mutually redundant features in Figure 7. We similarly see that increasing $\gamma$ increases the number of edges in graph $\mathcal{H}$, which correspondingly increases the number of mutually redundant features identified. For datasets with inherently low mutual redundancy, such as CIFAR10, this has the effect of reducing Post-hoc accuracy when $\gamma$ is increased past a certain threshold. Therefore in practice $\gamma$ should be selected either through cross validation, or by examining the density curve as in Figure 6.

### G.2.5 TIME COMPLEXITY DETAILS WITH UNIVARIATE COMPARISON.

Table 8 includes the full feature attribution timing results. Note that L2X requires an initial training stage for neural network-based explainer model, which is not included in these results. Once this explainer model is trained, the topk features are obtained through single forward pass, which is the activity measured in Table 8. All experiments are performed on an internal cluster equipped with Intel Gold 6132 CPUs. The evaluations on CIFAR10, MNIST, IMDB datasets were calculated using GPUs (Nvidia Tesla V100), whereas the other datasets were trained without a GPU. Finally, the calculated times were averaged over all samples used in the experiments (500 samples, unless the

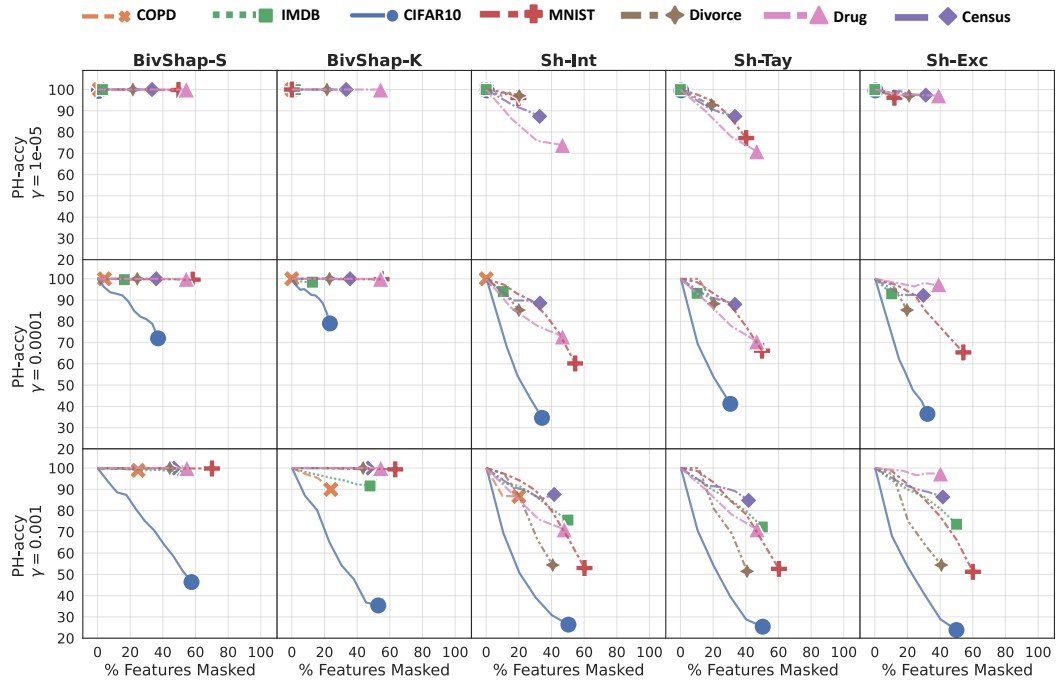

Figure 7: Posthoc Accuracy evaluated on Mutual Redundancy masking derived from graph $\mathcal{H}$. Strongly connected nodes in $\mathcal{H}$ are randomly masked with increasing mask sizes until a single node remains, represented by the final marker for each dataset. Each row represents a different selection of threshold parameter $\gamma$. Note that we cannot run Sh-Tay and Sh-Exc methods on the COPD dataset due to their computational issues with the large number of features.

| | | Bivariate Methods | | | | | Univariate Methods | | | GNN Methods |
|---|---|---|---|---|---|---|---|---|---|---|
| Dataset | # Features | Ours-SS | Ours-K | Sh-Int | Sh-Tay | Sh-Exc | Sh-Sam | kSHAP | L2X | GNNExp |
| COPD | 1077 | 5942 | 36 | 2877 | 112900* | 838200* | 3047 | 1.4 | 0.00 | 10.9 |
| CIFAR10 | 255 | 218 | 2.5 | 101 | 2819* | 6267* | 140 | 0.65 | 0.00 | 0.79 |
| MNIST | 196 | 116 | 1.5 | 48 | 1194* | 2350* | 57 | 0.34 | 0.00 | 0.42 |
| IMDB | ≤400 | 207 | 1.9 | 160 | 1279* | 1796* | 103 | 0.40 | 0.00 | 0.73 |
| Census | 12 | 2.7 | 0.20 | 2.6 | 11.6 | 5.3 | 1.6 | 0.83 | 0.00 | 0.17 |
| Divorce | 54 | 18.2 | 0.34 | 6.5 | 63.2 | 93.3 | 11.3 | 0.16 | 0.00 | 0.15 |
| Drug | 6 | 2.3 | 0.07 | 1.21 | 181 | 0.96 | 1.26 | 0.10 | 0.00 | 1.54 |

Table 8: Time comparison in seconds per data sample for the methods used for the post-hoc accuracy and AUC calculations. Fields indicated by * which were averaged over 5 samples due to computational cost, otherwise time calculations were averaged over 500 samples (or the total number of test samples if fewer than 500)

dataset has less than 500 samples total), except for the fields indicated by * which were averaged over 5 samples due to computational cost.

As previously mentioned in Sec G.1.1, the Bivariate Shapley method can be applied naively to any removal-based explanation method repeating the explainer's calculations $d$ times, where $d$ is the number of data features. It follows that our method's time complexity is dependent on the choice explanation method and, when implemented naively, linearly scales that method's complexity by the number of features. Certain methods, such as KernelSHAP, can be adapted to realize even more efficient implementations of Bivariate Shapley, which we outline in App G.1.1. We provide time comparisons to competing methods in Table 1.

| Group | Pathway Name | Genes | q-value |
|---|---|---|---|
| Smoker | **GSE25123_WT_VS_PPARG_KO_MACROPHAGE_IL4_STIM_DN** | **23** | **0.02** |
| | GSE32986_GMCSF_VS_GMCSF_AND_CURDLAN_LOWDOSE_STIM_DC_UP | 24 | 0.16 |
| | GSE45365_WT_VS_IFNAR_KO_CD11B_DC_UP | 18 | 0.20 |
| | GSE22886_NAIVE_VS_MEMORY_TCELL_DN | 32 | 0.20 |
| | GSE40274_FOXP3_VS_FOXP3_AND_LEF1_TRANSDUCED_ACTIVATED_CD4_TCELL_UP | 25 | 0.21 |
| | GSE32986_UNSTIM_VS_CURDLAN_LOWDOSE_STIM_DC_DN | 28 | 0.32 |
| NonSmoker | **GSE25123_WT_VS_PPARG_KO_MACROPHAGE_IL4_STIM_DN** | **23** | **0.01** |
| | GSE32986_UNSTIM_VS_CURDLAN_LOWDOSE_STIM_DC_DN | 28 | 0.13 |
| | GSE32986_GMCSF_VS_GMCSF_AND_CURDLAN_LOWDOSE_STIM_DC_UP | 24 | 0.21 |

Table 9: GO enrichment results for the redundancy-based ranking of graph $\mathcal{G}$ for Smoker and Nonsmoker subgroups. Gene pathways with q-value $< 0.05$ are bolded.

### G.2.6 GENE ONTOLOGY ENRICHMENT ANALYSIS OF COPDGENE DATASET

Chronic Obstructive Pulmonary Disease (COPD) is a chronic inflammatory lung disease. The relation between COPD and smoking is well-established; it has been shown that smoking increases the risk of developing lung disease through a variety of ways, such as increasing lung inflammation (Arnson et al., 2010). Here, we investigate the relation between gene expression data and smoking status in COPDGene data. We show the interpretation power of our methods by relating our most influential genes to biological pathways which correspond to smoking. We performed Gene Set Enrichment Analysis (GSEA) using the GenePattern web interface (Reich et al., 2006) on the ranking of influential features, which we generate as follows. We first calculate graph $\mathcal{G}$ locally, as in Alg. 1. We then create the global $\mathcal{G}$ graph for each subgroup, smokers and non-smokers, by averaging the $\mathcal{G}$ adjacency matrix over all samples within each subgroup (Fig. 8). We directly apply the ranking algorithm outlined in Section G.1.1 to obtain subgroup-specific importance scores. We use the list of 1,079 unique gene names with their associated importance score as input into the GenePattern interface. Gene set enrichment for these two groups was calculated using the GSEAPreranked module with 1000 permutations, using the Hallmark (h.all.v7.4.symbols.gmt) and Immunologic gene sets (c7.all.v7.4.symbols.gmt). We observed genetic pathways corresponding to Macrophages as statistically significant at a q-value $\leq 0.05$ (the pathway table is in the App., Table. 9). Macrophages are a type of immune cells that can initiate inflammation, and they also involve the detection and destruction of bacteria in the body. The relation between such cells and smoking has been observed in biological domain; many studies have observed that smoking induces changes in immune cell function in COPD patients (Yang & Chen, 2018; Strzelak et al., 2018).

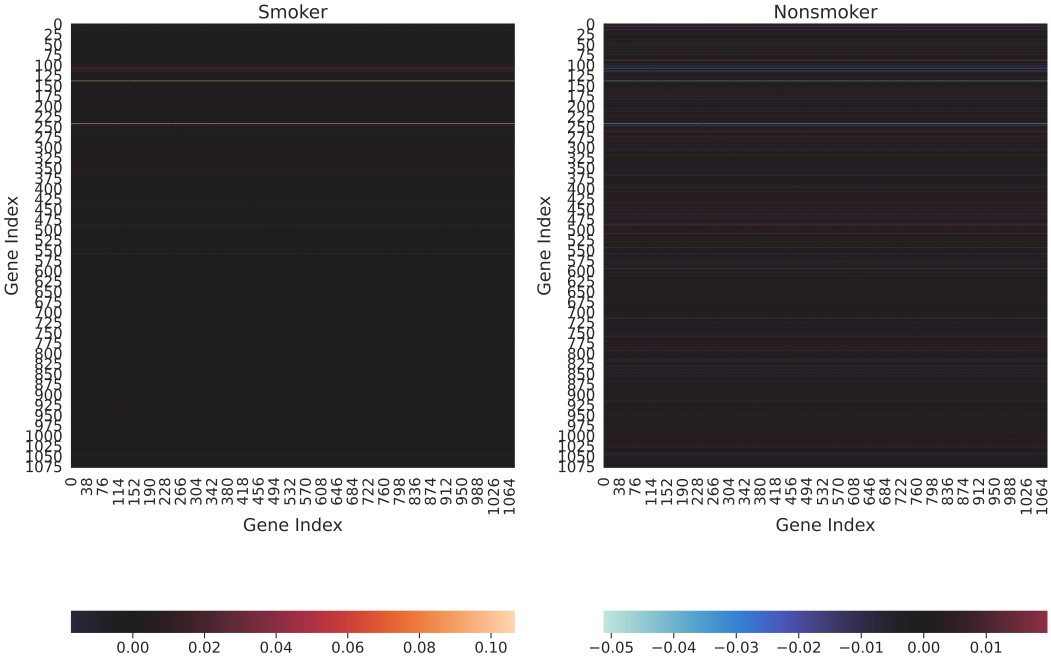

Figure 8: Adjacency matrix for graph $\mathcal{G}$, averaged over Smoker and Nonsmoker subgroups and displayed as a heatmap.

