# OpenReview forum: "Explanations of Black-Box Models based on Directional Feature Interactions"
_ICLR.cc/2022/Conference — ICLR 2022 Spotlight_

### Official Review · Reviewer_5zsk · 2021-10-25

**Correctness:** 4
**Technical Novelty And Significance:** 3
**Empirical Novelty And Significance:** 3
**Recommendation:** 8
**Confidence:** 3

**Main Review:**

Strengths

The proposed methodology studies a relevant and important problem in the area of feature-based explanations by identifying the influence and directional feature interactions for an individual model prediction.

As far as I can tell, the work is original and distinct from previous work on identifying influential feature interactions which assume symmetrical interactions between pairs of features.

The proposed method appears to be technically sound and general, providing a framework for computing directional bivariate-explanation maps using any univariate explainer.

Experiments are performed on 7 datasets with image, text, and tabular data using deep-learning and tree-based models; the paper also compares the proposed approach to an appropriate set of relevant baselines.

The evaluation does a nice job of illustrating and validating the ability of the proposed method to identify mutual and directional redundancies between features, a significant shortcoming of existing approaches.


Weaknesses

I'm a bit confused about the results in section "(3) Influential Feature Evaluation"; first, it is not clear to me what the objective of the experiment is, i.e., is it to maintain high accuracy as the least important features are removed (if so, perhaps this should be stated in the text or in Figure 3; also, why not remove the most influential features and observe which methods degrade classification accuracy the fastest as in previous work (Hooker et al. 2018))? In any case, this section suggests that BivShap-S performs better than all other methods, though in Figure 3 it looks as though BivShap-K performs best. Also, why are the univariate approaches--methods designed specifically to identify the most influential features--not performing just as well as the bivariate approaches on this task?

In Table 2, why is posthoc performance so high (often 99 or 100 percent) when masking sinks? I thought these are typically characterized as the most important features (e.g., "awful" or "worst" in Figure 1). Also, for COPD, CIFAR10, and IMDB, the percentage of sources masked in Table 2 is very high compared to the percentage of sinks masked; could the low predictive performance simply be a result of removing the majority of the information and not necessarily the identification of important features?

It seems the E^2 matrix (bivariate interactions) would become unwieldy as d (the number of features) increases, reducing the interpretability of any manual inspection of the resulting directional graph G. This problem is exacerbated via analysis of higher-order multivariate interactions.

How does the choice of the univariate explainer affect the bivariate explanations? Is there a significant difference when using one univariate explainer versus another?


Minor Weaknesses

Why is BivShap-K not included in Figure 2?

It's difficult to tell where some of the lines are in Figure 3, consider using some transparency when plotting. Also consider using vector graphics for plots to allow zooming in (when viewed online) without losing clarity.

Missing '.' at the end of sentence 4 in the section, "Illustrative Example".


Additional Questions

What is the utility function used in the experiments to compute approximate BivShap-S?

The "Data." section says a neural network and a tree-based model is trained on the Divorce dataset, is this right?

What are the hardware details used for the experiments?

**Summary Of The Paper:**

This paper attempts to alleviate the shortcomings of existing local-feature-interaction explainers that assume symmetrical feature interactions by proposing a bivariate feature-explanation map that can capture asymmetrical (directional) feature interactions; this analysis provides evidence of mutual and directional redundancy, offering a comprehensive understanding of the features most influential for a given prediction. This proposed approach can also be instantiated using any univariate feature-based explanation method. Empirical results on image, text, and tabular data show the ability of the proposed method to accurately identify mutual and directional redundancies.

**Summary Of The Review:**

The paper proposes a method for computing asymmetrical feature interactions for local explanations of model predictions. The proposed approach is better able to identify mutual and directional redundancies compared to existing methods on a wide range of datasets. Overall, the paper is well-written (aside from some minor clarity issues, see review above) and provides a method that can extend the use of any univariate explainer to generate directional bivariate feature-interaction explanations. One of my main concerns is how this bivariate explainer performs better than univariate explainers at a task seemingly designed for univariate explainers.

---

> ### Author Response · Authors · 2021-11-20
> **Part 1/2; Experiment objective in Figure 3; Performance of univariate in comparison with bivariate; Sink and source representation;**
>
> Thank you for reviewing and providing feedback on our submission. We appreciate the detailed notes on our experiment results and are encouraged that you recognize the value of mutual and directional redundancies.
>
> ---
> **Q: objective of the experiment [in section (3)]? also, why not remove the most influential features and observe which methods degrade classification accuracy the fastest as in previous work (Hooker et al. 2018))?**
>
> That is correct, the objective is to maintain high accuracy as the least important features are removed. We have also evaluated performance by removing the most important features first through deletion AUC (dAUC) [1]. The results are shown in Table 4, App. F.2.1 on page 24. dAUC measures the change in model output as features are iteratively removed, summarized by area under the curve.
>
> ---
> **Q: why are the univariate approaches … not performing just as well as the bivariate approaches on this task?**
>
> * In Fig 3, Sh-Sam and kSHAP are partially covered by the BivShap-K curves; we have revised Fig 3 to make those curves more visible.  Thank you for your suggestions of using transparency and vector graphics.
> * As seen in Fig 3, univariate Shapley methods Sh-Sam and kSHAP perform almost equally in comparison with BivShap-K except in the case of CIFAR10. So in most examples, univariate Shapley is performing as well as BivShap-K.
>
>
> * In the case of CIFAR10, we hypothesize that capturing redundancy between features enables the model to remove features more effectively. To illustrate, under univariate Shapley if a feature has a high Shapley value then it is important and should not be removed. Yet if that same important feature is redundant with respect to some other feature, then using PageRank on the graph $\mathcal{G}$ will result in a lower steady-state for the first feature, as its information is already captured elsewhere. Therefore the first feature would be removed sooner in our experiment without any loss of accuracy despite having a high shapley value. This relation cannot be captured by univariate Shapley due to its inability to capture redundancy.
>
>
> ---
>
> **Q: In Table 2, why is posthoc performance so high … when masking sinks? I thought these are typically characterized as the most important features?**
>
>
>
> As you pointed out, sinks represent important features for Graph $\mathcal{G}$. However, in graph $\mathcal{H}$, which is defined as the complement of the support of graph $\mathcal{G}$, sinks represent the most redundant features. So the notion of sink and source in graph $\mathcal{G}$ flips in graph $\mathcal{H}$, i.e., in graph $\mathcal{H}$, source represents the set of features that makes everything else redundant, and sink represents the most redundant features. This definition is aligned with the results in table 2; i.e, masking $\mathcal{H}$-sinks keeps the accuracy intact while removing $\mathcal{H}$-sources leads to a loss of accuracy.
>
>
> To improve clarity of difference of sinks and sources between two graphs , we have added figure 5 in App. F.1.1 on page 20, which compares graph $\mathcal{G}$ and graph $\mathcal{H}$ for the IMDB example.
>
>
>
>
> ---
> **Q: Also, for COPD, CIFAR10, and IMDB, the percentage of sources masked in Table 2 is very high compared to the percentage of sinks masked; could the low predictive performance simply be a result of removing the majority of the information...?**
>
> This suggests that COPD, CIFAR 10, and IMDB contain many features with unique information (sources). When we remove the identified sources, the combined loss of unique information results in a large reduction of post-hoc accuracy. Conversely, when removing sinks (redundancies), we expect a minimal loss of information, and hence a minimal impact to post-hoc accuracy.
>
> ---
> **Q: How does the choice of the univariate explainer affect the bivariate explanations? Is there a significant difference when using one univariate explainer versus another?**
>
> Indeed, different univariate explainers lead to different bivariate explanations. For instance, using L2X [2], which uses instance-wise feature selection, would lead to unweighted directed graphs. In contrast, Bivariate Shapley leads to a weighted directed graph $\mathcal{G}$. Our focus on exploring SHAP explainers is two-fold. First, Lundberg et al [3] unified 6 other univariate explainers under SHAP and therefore our bivariate extension also would unify the other 6 bivariate extensions. Second, using Shapley formalisms, the redundancy patterns follow almost transitivity (Theorem 1), which is aligned with the intuition of humans for redundancy patterns.
>
> ---

---

> > ### Author Response · Authors · 2021-11-20
> > **Part 2/2; Additional Questions**
> >
> > **Q: Additional Questions and Minor Weaknesses**
> >
> > * In BivShap-S the utility function is $u(S) = \mathbb{E} _{w\sim \mathcal{B}}[P(Y=\hat y |X = x _S \cup w _{\bar S})]$, where $\hat{y}$ is the model's predicted class, $\bar{S}$ is the complement of $S$, and $w$ represents samples drawn from a baseline distribution $\mathcal{B}$.
> > * Thank you for pointing out the typo in the “Data” section. We train a 3-layer neural network on the Divorce Dataset. The second mention of “Divorce” is intended to be the Drug dataset. This is now fixed in the current revision.
> > * Regarding hardware details: experiments were evaluated on an internal cluster equipped with Intel Xeon Gold 6132 CPUs and Nvidia Tesla V100 GPUs. We have moved the details from the appendix to the main paper for clarity.
> >
> > * Regarding the BivShap-K comparison in Figure 2: we omitted this plot to save space since the results are similar to BivShap-S, however we have now included the comparison.
> >
> > ---
> > [1] Petsiuk et al. “RISE: Randomized Input Sampling for Explanation of Black-Box Models.”
> >
> > [2] Jianbo  Chen et al. Learning  to  explain:   An information-theoretic perspective on model interpretation.
> >
> > [3] Lundberg et al. “A Unified Approach to Interpreting Model Predictions.”

---

> > > ### Comment · Reviewer_5zsk · 2021-11-22
> > > **Response**
> > >
> > > I have read the author response and intend to increase my score to 8. I do encourage the authors to clarify the experimental objective regarding Figure 3.

---

> > > > ### Author Response · Authors · 2021-11-22
> > > > **Response**
> > > >
> > > > Thank you for your consideration. We appreciate your feedback and have revised the paper to include the experiment objective in the Figure 3 caption.

---

### Official Review · Reviewer_GZtH · 2021-11-01

**Correctness:** 4
**Technical Novelty And Significance:** 3
**Empirical Novelty And Significance:** 3
**Recommendation:** 8
**Confidence:** 3

**Main Review:**

Strength
1. The paper proposed an interesting and innovative graph-based method to generalize the univariate Shapley value method. The natural application of graph algorithms and concepts like PageRank, connected components is very interesting.
2. The authors carry out an extensive experiment on a diverse set of models and tasks with comparison to several state-of-the art method.
3. The empirical evaluation is quite convincing as the authors compare the performance of the model to tasks of bivariate but also univariate like feature importance. The proposed method performed better in both types of tasks.
4. The authors provide both theoretical justification as well as good illustrative examples.

Weakness
1. One important aspect is that there is quite some noise in the graph edge from Shapley estimation. It would be interesting to mention how the threshold can effect the results of the methods.


**Summary Of The Paper:**

In this paper, the authors generalize the univariate Shapley method to bivariate Shapley method. The authors first build a directly graph based on the asymmetric bivariate Shapley value (adding feature j to all sets contained feature i). Then several graph algorithms are applied to analyze the directly graph to derive (1) univariate feature importance available in univariate approach and (2) relations like mutually redundancy only available in bivariate approaches. Experiments on several datasets with comparison to existing methods demonstrated the superiority of the proposed method.

**Summary Of The Review:**

The paper proposed an interesting and innovative solution to an important problem. Also the evaluation is very convincing.

---

> ### Author Response · Authors · 2021-11-20
> **On the effects of different thresholding on redundancy patterns**
>
> We thank the reviewer for taking the time to provide feedback on our work. We are encouraged to know that you appreciate our experiments on the diverse set of models and datasets, as well as the theoretical justifications for Bivariate Shapley. Below we address your comment on the effects of thresholding; please let us know if you have any more questions.
>
> ---
>
> **Q: One important aspect is that there is quite some noise in the graph edge from Shapley estimation. It would be interesting to mention how the threshold can effect the results of the methods.**
>
> As you mentioned, the choice of threshold parameter $\gamma$ in calculating the $\mathcal{H}$-graph has an impact in the discovery of mutually-redundant and directionally-redundant features. We investigate the choice of $\gamma$ in Appendix F.2.4., page 25, by evaluating the change in average graph density of the data samples as $\gamma$ increases and its effect on Post-Hoc accuracy after masking directionally-redundant features (Figure 6). We have also added additional results in Figure 7 to explore how increasing $\gamma$ affects the identification of mutually redundant features. We similarly see that increasing $\gamma$ increases the number of edges in graph $\mathcal{H}$, which also increases the sensitivity (true positive rate) of detecting mutually redundant features with the tradeoff of reduced specificity (true negative rate) . For datasets with inherently low mutual redundancy, such as CIFAR10, this has the effect of reducing Post-hoc accuracy when $\gamma$ is increased past a certain threshold. Therefore in practice $\gamma$ should be selected either through cross validation, or by examining the density curve as in Figure 6.

---

> > ### Comment · Reviewer_GZtH · 2021-11-20
> > **Response**
> >
> > I have read the author response and thanks the authors for the answer.

---

### Official Review · Reviewer_k52n · 2021-11-02

**Correctness:** 3
**Technical Novelty And Significance:** 3
**Empirical Novelty And Significance:** 3
**Recommendation:** 8
**Confidence:** 4

**Main Review:**

## Strength:

-   The paper is well organized. The authors provide sufficient background and motivation. Despite the fact that the Shapley value has been applied to a wide variety of AI applications, there is precisely no method to explore how the features affect each other non-symmetrically.
This work takes an interesting approach. Making the interaction matrix conditional on a particular feature by modifying the original Shapley value sounds natural and simple.


-   The paper provides extensive empirical results, which indicate that the approach is a good explanation for measuring the influence of features on each other.


## Weakness:

-   I am concerned about how well the sampling and kernel methods estimate the true bivariate Shapley explanation map?
 What is the variance of the estimation, and, how about the performance if one uses a true explanation map? In figure 2 and table 2, the authors only show BivShap-S and not the BivShap-K which is used in all the other experiments. How does the performance look like with BivShap-K.

- The bivariate Shapley explanation in  Eq (6) has a conditional interpretation as the authors describe, specifically, $E^2(u)_{ij}$ represents the importance of feature j conditioned on feature i being present.  Could this conditional representation be rigorously verified, say, in the EBM (energy based model) treatment of valuation criteria in [1]:

- The monotone utility function is a necessary condition of the transitivity property of mutual redundancy in Corollary 1.1. Does the model used in experiments satisfy the monotone condition? If not, how do you select the features to mask in the mutual redundancy experiment of figure 2?

-   The authors claim that their method can be extended into high-order scenarios, and provide a method to calculate the multivariate explanation map in appendix E. As far as I see, a more complicated problem is how to define the interaction types in high-order scenarios and how to discover them? Does the PageRank algorithm fit into this case? I would appreciate a comment from the authors providing more discussions about it. It will be exactly encouraging for future works.

-   The part of experiments is hastily written and needs more refinement: algorithm 1 and algorithm 2 are totally the same; in F.1.1, Fig. 2(a) and 2(b) are missing; the reference of table and figure are interleaved in the main paper and appendix, that is quite unfriendly to readers. It is also not well illustrated how the PageRank algorithm is used to search for influential and redundant features. I think it is worthy of using an additional section to illustrate them.


[1] Bian, Y., Rong, Y., Xu, T., Wu, J., Krause, A., & Huang, J. (2021). Energy-Based Learning for Cooperative Games, with Applications to  Valuation Problems in Machine Learning. arXiv preprint arXiv:2106.02938.


**Summary Of The Paper:**

In previous studies, Shapley value has been widely used to explain model predictions, in which previous studies generally characterized importance for each feature instance separately. While some works have explored the effect of combinatorial features on prediction, no works have explored how features interact to influence each other, such as whether feature A is redundant if feature B is present.

In order to achieve this goal, this paper offers an intuitive solution. The degree of feature interaction is estimated using a matrix called a bivariate Shapley explanation map. The j-th column and i-th row of the matrix represents how feature i influences the prediction if feature j has been included. Item ij can be seen as a revised Shapley value: rather than summing up the marginal contributions of all coalitions, it only considers them when feature j appears. They then propose four different types of interactions based on this definition: least/most influential features, directional/mutual redundancy. The experimental results indicate that the proposed method can efficiently discover these four types of feature interactions.


**Summary Of The Review:**

The idea and presentation of this work are, in my opinion, good despite existing problems.
Furthermore, I think it would be better to move the analysis of COPDGene data into the appendix since all experimental results in this section are missing from the main paper. The illustrative samples in the appendix should also be included as part of the main paper.

---

> ### Author Response · Authors · 2021-11-20
> **Variance of explanation map; Monotonicity condition; Higher order explanation**
>
>
> Thank you for taking the time to review our submission and provide feedback. We appreciate your comments on the novelty of our work, as well as your recommendation of the EBM paper from Bian et al which indeed seems to have a theoretical connection to our work. Below we address your concerns and outline the revisions we have made in response to your feedback.
>
> ---
> **Q: I am concerned about how well the sampling and kernel methods estimate the true bivariate Shapley explanation map? What is the variance of the estimation, and, how about the performance if one uses a true explanation map?**
>
> Note that calculating the Bivariate Shapley explanation map is equal to calculating $d$ univariate Shapley explanation maps. More specifically, each column of the $d \times d$ explanation map is a separate univariate explanation with utility function defined in Eq. 5 on page 4. Therefore our method can be generalized to a number of different univariate explanation techniques and approximation methods (see Sec. 3). As such, if we use an approximation method to estimate the explanation map, then the variance of the approximation in each column of Bivariate Shapley is the same as the variance of the respective univariate approximation method.
> For the specific cases of BivShap-S and BivShap-K, which extend the Shapley Sampling and KernelSHAP methods respectively, we can refer to prior works on estimating their variance. Štrumbelj and Kononenko [1] show that the variance of the Shapley Sampling estimation is $\frac{\sigma_i}{m}$, where $\sigma_i$ is the population variance of feature $i$ and $m$ is the number of Monte Carlo samples. Regarding KernelSHAP, a recent paper from Covert and Lee [2] argue that the variance KernelSHAP is difficult to characterize theoretically, but they explore its variance empirically in section 4.1 and introduce variance reduction techniques.
>
> ---
> **Q: In figure 2 and table 2, the authors only show BivShap-S and not the BivShap-K**
>
> The BivShap-K results were omitted due to space constraints since the results are similar to BivShap-S, however we have now included this comparison to Figure 2. We have also added the BivShap-K results for Table 2 to the appendix App. F.2.3. on page 25 (Table 6). Thank you for the feedback.
>
> ---
> **Q: Does the model used in experiments satisfy the monotone condition?**
>
> In Corollary 1.1, monotonicity of utility function is a sufficient condition for transitivity. This transitivity indicates that every strongly connected components of Graph $\mathcal{H}$ is a maximal clique. However, Theorem 1 represents an approximation to transitivity which we call “almost transitivity” in the case of non-monotone utility functions. In our experiments, we use the utility function $u(S) = \mathbb{E} _{w\sim \mathcal{B}}[P(Y=\hat y |X = x _S \cup w _{\bar S})]$, where $\hat{y}$ is the model's predicted class, $\bar{S}$ is the complement of $S$, and $w$ represents samples drawn from a baseline distribution $\mathcal{B}$. This utility is not monotone. Hence, we identify mutual redundancy via the approximation case in Theorem 1; i.e. by finding the quotient graph using the strongly connected components rather than the maximal cliques. The results from experiments in Figure 2 validate this approximation case: removing elements from the strongly connected components has minimal effect on post-hoc accuracy.
>
>
> ---
> **Q: How to find interaction in higher-order explanation maps?**
>
> Thank you for highlighting that our formulation can be extended to higher scenarios as we mentioned in the appendix E. Finding the important nodes in a hypergraph can still be done via pagerank for hypergraphs (e.g. [3], [4]). But a proper definition of mutual redundancy and directional redundancy would lead to higher order redundancies that, rather than focusing on pairwise redundancies, will generalize to subset vs subset redundancies. This topic does require further investigation.
>
> ---
> **Q: Small Fixes and Clarification in Experiment Section**
>
> * Thank you for pointing out these typos, we have fixed these issues in the current revision.
> * Thanks for pointing out this issue with clarity regarding the PageRank algorithm. In our paper we have described the three algorithms that identify: 1) Mutually Redundant Features, 2) Directionally Redundant Features, and 3) Influential Features. We use the PageRank algorithm in tasks 2 and 3, which we outline in the App. F.1.1 on page 18. We agree that the clarity of the algorithms could be improved; in the current revised version we have added the related pseudocode for each task in Alg. 3, Alg. 4, and Alg. 5. in App. F.1.1.
> ---
> [1] Erik Štrumbelj and Igor Kononenko. "Explaining prediction models and individual predictions with feature contributions."
>
> [2] Covert et al.. “Improving KernelSHAP: Practical Shapley Value Estimation via Linear Regression.”
>
> [3] Takai et al. “Hypergraph Clustering Based on PageRank.”
>
> [4] Tran, Loc et al. “PageRank Algorithm for Directed Hypergraph.”

---

> > ### Comment · Reviewer_k52n · 2021-11-22
> > **Thanks for the response**
> >
> > I have read the responses of the author, and also read the questions&answers of the other reviewers.
> > Now I intend to increase my score.

---

> ### Author Response · Authors · 2021-11-20
> **Relation with EBM paper**
>
>
> **Q: Could this conditional representation be rigorously verified, say, in the EBM (energy based model) treatment of valuation criteria in [1]**
>
> Thank you for noticing that this has a conditional interpretation, where it represents the importance of feature j conditioned on feature i.  We appreciate you pointing out this interesting concurrent paper that connects energy-based models to shapley in the univariate sense (and symmetric interactions). It would be interesting to explore this in future work and how it can be rigorously verified. But as a side note on the matter, we can say the following:
>    The $E^{2}(u) _{ij}$ formulation in eq(6) can be interpreted as a shapley value when player (feature) $i$ only plays the game only if player $j$ is already in the game. This means that $E^{2}(u) _{ij}$ is basically a filtered version of $E(u) _{i}$ where the game that player j is not in has been removed. To connect this to the EBM paper, we need to look at their interesting proof for Shapley value, i.e., they defined the evidence lower bound (ELBO) to be as follows:
>
> $$
> f_{mt}^{F}(x) =  \sum_{S\subseteq N} a(S) \prod_{i\in S}x_{i}
> $$
>
> Where $a(S) := \sum_{T\subseteq S} (-1)^{|S|-|T|} F(T)$, using this definition they proved shapley value has the following from:
>
> $$	Sh_{S} = \int_{0}^{1} \Delta_{S}f^{F}_{mt}(x1)dx		$$
>
> letting $S = \{i\}$ we have the univariate shapley value as:
>
> \begin{equation}
>     \begin{split}
>      Sh_{i} =  \sum_{i\subseteq S} \frac{1}{|S|}a(S) =  \sum_{i\subseteq S} \frac{1}{|S|} \sum_{T\subseteq S} (-1)^{|S|-|T|} F(T)
>     \end{split}
> \end{equation}
> As we said earlier for $Sh _{ij} := E^{2} _{ij} $ we filter elements in $Sh _{i} = E _{i}$ that don't have $j\in T$, and the rest of elements would appear exactly the same with the same coefficients as Shapley, hence as a consequence it seems to be the case that:
>
> \begin{equation}
>     \begin{split}
>      Sh_{ij} =  \sum_{i\subseteq S} \frac{1}{|S|}a(S,j) =  \sum_{i\subseteq S} \frac{1}{|S|} \sum_{j\in T\subseteq S} (-1)^{|S|-|T|} F(T)
>     \end{split}
> \end{equation}
>
> where $a(S,j) :=  \sum_{j\in T\subseteq S} (-1)^{|S|-|T|} F(T)$, note that if $j\notin S$ then summing over the empty set is $a(S,j) = 0$. The idea behind this equation can be seen as filtering all the $F(T)$ that $j\notin T$, and keeping the rest the same coefficients. This is the same idea behind $E^{2}(u)_{ij}$ that it filters part of the Shapley value that $j\notin T$.
>
>  Hence, computing the Bivariate Shapley value for feature $i$ will have the following form:
>
> $$	E^2_{ij} = Sh_{ij} =  \int_{0}^{1} \Delta_{\{i\}}f^{F}_{mt}(x1,j)dx	$$
>
>
> where $$f_{mt}^{F}(x,j) = \sum_{S\subseteq N} a(S,j) \prod_{i\in S}x_{i}$$
>
>
> It does seem that $E^2_{ij}$ has a similar connection as of Shapley interaction index and Shapley value. This could be an interesting future direction to investigate.

---

### Official Review · Reviewer_WTCZ · 2021-11-03

**Correctness:** 4
**Technical Novelty And Significance:** 3
**Empirical Novelty And Significance:** 2
**Recommendation:** 8
**Confidence:** 4

**Main Review:**

+ The studied problem is very important and interesting. XAI is highly related to the model's trust and safety

+ The proposed method- using Bivariate Shapley values to study feature interactions- is novel and reasonable. Considering data in DAG format is reasonable and their connections are represented as edges.

+ Experimental results are very promising and the analysis is very useful.

+ It is great to see some discussion about the connections between the proposed method and GNN explanation techniques.

- While this work discussed several GNN explanation techniques, it would be better to compare the proposed method with them. I believe this can be done with simple modifications. For example, GNNExplainer (also some other GNN explanation techniques) does not require the model to be GNN---actually, it also treats the model as the black-boxes. Then when the input is represented as the DAG and the same feature interactions can be captured. Such experimental comparisons can better connect the proposed method to the GNN explanation area.

- A better literature review of GNN explanation techniques is desired.

**Summary Of The Paper:**

This work proposed to study the directional feature interactions to explain deep models. The proposed method is a graph-based explainer and the data can be considered as graphs. Then it studies the Bivariate Shapley values to consider the directional feature interactions. Experiments on several datasets show very promising results.

**Summary Of The Review:**

Overall, I think the studied problem is important and the proposed method is novel and reasonable. Considering that some experimental comparisons are missing, I believe this is a borderline paper and recommend a weak accept.

---

> ### Author Response · Authors · 2021-11-19
> **Comparison with GNNExplainer and additional literature review on GNN explanation techniques**
>
> We would first like to thank the reviewer for their time and the helpful feedback. We are encouraged that you see the importance of feature interactions in XAI and we acknowledge the relevance of the GNN explanation techniques you mention. Below we outline the revisions we have made based on your feedback.
>
> ---
>
> **Q: Comparison with GNN explanation techniques**
>
> Thanks for pointing out this comparison. As you suggested, we implemented GNNExplainer [1] using the Pytorch Geometric package [2]. We defined the input to GNNExplainer as a complete graph with nodes defined as the features of the data sample. This explanation technique was applied to the same black-box models we used in our experiments section; as you pointed out, these models do not need to be GNNs. We have updated Figure 3 to include this comparison. The full details of our GNNExplainer implementation are listed in App. F.1.2 on page 22.
>
> We can see that while GNNExplainer does produce graph-based explanations, the resulting explanation does not perform as well as Bivariate Shapley in terms of identifying redundancies. In particular, we observe that the Post-Hoc accuracy shown in Figure 3 is significantly lower than that of the Bivariate Shapley explanations. In addition, when using the explanations from GNNExplainer to find Mutually Redundant features as in Figure 2, we observe that the explanations do not find any such redundancies and thus no features are masked.
>
> We hypothesize that one contributing factor in its lower performance is that GNNExplainer and similar masking-based methods rely on masking graph edges, nodes, and/or node features to build their explanation. Since the black-box models we are explaining are not GNNs, they do not use any information from the graph structure (e.g. edges) in their prediction, therefore masking the edges of the complete graph inputs will not have any effect on the model prediction.
>
> Thank you again for bringing this to our attention; we believe that this comparison adds value to our paper. The related experiment code has also been uploaded to the anonymized repository.
>
> ---
>
>
> **Q: Additional literature review on GNN explanation techniques**
>
> Thanks for the feedback, we have expanded the literature review on page 2 to include more detail regarding GNN-based explainers and their applicability to non-GNN black-box models.
>
> ---
>
> [1] Rex Ying, Dylan Bourgeois, Jiaxuan You, Marinka Zitnik, and Jure Leskovec. GNNExplainer:
> Generating explanations for graph neural networks. Advances in neural information processing
> systems, 32:9240, 2019.
>
> [2] Fey, M., & Lenssen, J. E. (2019). Fast Graph Representation Learning with PyTorch Geometric (Version 2.0.2) [Computer software]. https://github.com/pyg-team/pytorch_geometric

---

> > ### Comment · Reviewer_WTCZ · 2021-11-20
> > **Thanks for the author response**
> >
> > I have read the author response and I believe my concerns are largely addressed. I am increasing my score to 8.

---

### Decision · Program_Chairs · 2022-01-20

**Decision:**

Accept (Spotlight)

**Comment:**

In this paper, the authors generalize the univariate Shapley method to bivariate Shapley method. The authors first build a directly graph based on the asymmetric bivariate Shapley value (adding feature j to all sets contained feature i). Then several graph algorithms are applied to analyze the directly graph to derive (1) univariate feature importance available in univariate approach and (2) relations like mutually redundancy only available in bivariate approaches. Experiments on several datasets with comparison to existing methods demonstrated the superiority of the proposed method.  All reviews are positive.